# Pleiotropy increases parallel selection signatures during adaptation from standing genetic variation

**Wei-Yun Lai[1,2], Sheng-Kai Hsu[1,2], Andreas Futschik[3], Christian Schlötterer[1]\***

[1]Institut für Populationsgenetik, Vetmeduni Vienna, Vienna, Austria; [2]Vienna Graduate School of Population Genetics, Vetmeduni Vienna, Vienna, Austria; [3]Department of Applied Statistics, Johannes Kepler University Linz, Linz, Austria

## eLife Assessment

This study makes the **important** finding that pleiotropy is positively associated with parallelism of evolutionary responses in gene expression. This finding, if true, runs counter to current expectations in the field. The analysis uses state-of-the art experimental evolution approach to study the genetic basis of adaptation of *Drosophila* simulans to a hot environment. Although the experimental results are **convincing**, the theoretical model is **incomplete**, due to several unusual assumptions. It remains to be seen whether the main conclusion can be replicated in other contexts.

**\*For correspondence:**
schlotc@gmail.com

**Competing interest:** The authors declare that no competing interests exist.

**Abstract** The phenomenon of parallel evolution, whereby similar genomic and phenotypic changes occur across replicated pairs of populations or species, is widely studied. Nevertheless, the determining factors of parallel evolution remain poorly understood. Theoretical studies have proposed that pleiotropy, the influence of a single gene on multiple traits, is an important factor. In order to gain a deeper insight into the role of pleiotropy for parallel evolution from standing genetic variation, we characterized the interplay between parallelism, polymorphism, and pleiotropy. The present study examined the parallel gene expression evolution in 10 replicated populations of *Drosophila simulans*, which were adapted from standing variation to the same new temperature regime. The data demonstrate that the parallel evolution of gene expression from standing genetic variation is positively correlated with the strength of pleiotropic effects. The ancestral variation in gene expression is, however, negatively correlated with parallelism. Given that pleiotropy is also negatively correlated with gene expression variation, we conducted a causal analysis to distinguish cause and correlation and evaluate the role of pleiotropy. The causal analysis indicated that both direct (causative) and indirect (correlational) effects of pleiotropy contribute to parallel evolution. The indirect effect is mediated by historic selective constraint in response to pleiotropy. This results in parallel selection responses due to the reduced standing variation of pleiotropic genes. The direct effect of pleiotropy is likely to reflect a genetic correlation among adaptive traits, which in turn gives rise to synergistic effects and higher parallelism.

## Introduction

Distinguishing selection from stochastic changes continues to be a major challenge in evolutionary research. The evidence for selection mostly comes from parallel evolution, where similar genes, traits, or functions are identified in replicate populations exposed to similar environments (***Colosimo et al., 2005***; ***Westram et al., 2021***; ***van der Zee et al., 2022***; ***Zong et al., 2021***). Given that non-parallel evolution does not preclude the possibility of selection (***Barghi et al., 2019***; ***Barghi et al., 2020***),

there has been a sustained interest in elucidating the factors that influence the extent of genetic parallelism (i.e. how similar is the response of the same gene to selection in multiple populations from the same/similar ecological niche(s)) (*Bolnick et al., 2018*).

It has been proposed that the degree of pleiotropy, the number of traits affected by a single gene, maybe a potential factor associated with parallelism (*Bolnick et al., 2018*; *Conte et al., 2015*; *Rennison and Peichel, 2022*; *Stern, 2013*). The majority of hypotheses regarding the influence of pleiotropy on parallel selection responses are (implicitly) related to the ideas of *Fisher, 1930*, later modified by *Orr, 2000*, focusing on the cost of pleiotropy. This concept posits that any new mutation in a pleiotropic gene results in a shift in trait values away from their optimum across multiple traits. New mutations in highly pleiotropic genes are more likely to have negative fitness effects than mutations in less pleiotropic genes as more traits are displaced from their optima. This 'cost of pleiotropy' model (*Orr, 2000*) is evidenced by the lack of sequence divergence observed in genes with a higher level of pleiotropy (*Fraser et al., 2002*; *Hahn and Kern, 2005*; *Masalia et al., 2017*; *Josephs et al., 2017*). Consequently, it can be postulated that less pleiotropic genes are more likely to contribute to adaptation over long evolutionary timescales, given that they have a higher probability of accumulating beneficial genetic variants. This results in a more parallel selection response in different populations/species (*Hansen, 2003*).

Fisher's model of pleiotropy is simplistic in its assumption of uncorrelated traits and uncorrelated selection. It has been proposed that the influence of pleiotropy on the selection response strongly depends on the genetic correlation of the affected traits and the underlying fitness function which connects phenotypic values with fitness (*Blows, 2007*; *Lande and Arnold, 1983*; *Lande, 1979*). If the fitness effects of the affected traits are positively correlated (synergistic pleiotropy), this may result in stronger net selection (*Thorhölludottir et al., 2023*) and consequently in more parallel selection signatures (*MacPherson and Nuismer, 2017*). This implies that the correlation of fitness effects determines whether pleiotropy increases or decreases parallel selection signatures.

The impact of pleiotropy on parallel evolution has typically been considered in the context of long-term adaptation via de novo mutations. However, many studies of parallel evolution have focused on short-term adaptation [e.g. *Rivas-Sánchez et al., 2023*; *Huang et al., 2021*; *Heckley et al., 2022*; *Fischer et al., 2021*; *Elmer et al., 2014*; *Butlin et al., 2014*; *Hsu et al., 2021*; *Otte et al., 2021*], where standing genetic variation plays a more significant role in selection responses than de novo mutations. Nevertheless, it remains unclear whether a similar impact of pleiotropy on evolutionary parallelism is to be expected for standing genetic variation.

Indeed, the empirical evidence regarding the role of pleiotropy for intra-specific adaptive responses is somewhat inconclusive. A gene expression study in European grayling populations adapted to different temperature regimes provided compelling evidence for the constraining effect of pleiotropy (*Papakostas et al., 2014*), which supports the cost of complexity concept. A comparison of gene expression changes in natural *Drosophila* populations along a temperature cline and populations evolved in the laboratory to different temperature regimes demonstrated the importance of correlated fitness effects (*Thorhölludottir et al., 2023*). A direct test of the influence of pleiotropy on parallel evolution was conducted on 19 independent contrasts of stickleback populations (*Rennison and Peichel, 2022*). The authors discovered that pleiotropic genes were more likely to exhibit parallel adaptation signatures (*Rennison and Peichel, 2022*), thereby supporting the importance of synergistic pleiotropy.

Previous studies on pleiotropy and parallelism did not account for polymorphism (standing genetic variation), which plays a central role in selection responses during short-term adaptation. Although natural variation provides the basis for adaptive responses, it is not well-documented how ancestral variation (polymorphism) determines the level of genetic parallelism during short-term adaptation in replicate populations. Moreover, polymorphism of a gene is also affected by its pleiotropic level while the expectations regarding the influence of pleiotropy on the extent of polymorphism varies. The cost of complexity predicts that pleiotropic genes with uncorrelated fitness effects will be subject to stronger purifying selection, which will result in reduced polymorphism. In the case of multiple selected traits with aligned fitness effects, synergistic pleiotropy may facilitate the occurrence of selective sweeps, and thereby reducing variation in pleiotropic genes. Alternatively, it may be possible that once pleiotropic alleles have been established in a population, their multivariate fitness effects lead to a balanced polymorphism, which in turn maintains higher levels of variation. Empirical studies have

observed a negative correlation between the level of pleiotropy and sequence/expression polymorphism (*Lemos et al., 2004*; *Mähler et al., 2017*). This suggests that pleiotropy may preferentially reduce variation, either by purifying selection, synergistic effects, or a combination of both.

In this study, we employed gene expression analysis to investigate the interplay between polymorphism and pleiotropy on the degree of parallelism of adaptive responses. We assessed the selective response by measuring gene expression differences between populations from different environments, a widely employed approach [e.g. *Nourmohammad et al., 2017*; *Hsu et al., 2020*; *Yu et al., 2023*; *Velotta et al., 2017*; *Tang et al., 2017*; *O'Neil et al., 2014*; *Margres et al., 2017*]. Specifically, we compared gene expression in the ancestral *D. simulans* population with that of flies that had evolved for 100 generations in a novel hot temperature regime. To infer parallel evolution, the gene expression change was measured in 10 replicate populations. By focusing on the evolutionary response, we circumvent the challenge of connecting the phenotype to an unknown fitness landscape. Furthermore, we take advantage of well-established pleiotropy estimates (*McShea, 2000*; *Proulx et al., 2005*; *He and Zhang, 2006*; *Mank et al., 2008*) to evaluate the influence of pleiotropy on parallel gene expression evolution. The natural variation in gene expression was determined by examining the differences in expression levels among individual flies from the ancestral *D. simulans* population.

We observed significant interplays between pleiotropy, ancestral variation, and parallel selection responses among genes. The parallelism of gene expression evolution is positively associated with pleiotropy but negatively associated with the ancestral variation in gene expression. Given that pleiotropy was also negatively associated with ancestral variation, it is crucial to differentiate between statistical causality and correlation. Causal analysis revealed that pleiotropy affects parallel evolution both directly and indirectly via ancestral variation. It is likely that the indirect effects operate via the reduction of ancestral variation in pleiotropic genes. The direct effects of pleiotropy on parallelism are probably best explained by synergistic pleiotropy which results in a stronger selection response.

## Methods
### Estimating gene pleiotropy

We approximated the pleiotropy of each gene with two alternative estimators, network connectivity, and tissue specificity ($\tau$).

$\tau$ indicates how specific the expression of a gene is across different tissues. $\tau$ was estimated for each gene using the adult male expression profiles on flyatlas2 (*Leader et al., 2018*) as: $\tau = \frac{\sum_i \left[ 1 - \frac{gene\ expression_i}{gene\ expression_{max}} \right]}{N-1}$, where $N$ is the number of tissues examined and $i$ indicates each of them (*Dean and Mank, 2016*). If a gene is only expressed in one tissue, $\tau$ will equal to 1 while it equals 0 when a gene is expressed at the same level across all tissues. The relationship between $\tau$ and pleiotropy is based on the idea that the genes expressed in many tissues are more likely to affect multiple traits than genes expressed in fewer tissues (*McShea, 2000*; *Mank et al., 2008*; *Allen et al., 2018*; *Watanabe et al., 2019*). Hence, we used 1-$\tau$ to indicate the pleiotropic effect of a gene.

Network connectivity has also been used as a proxy for pleiotropy, as more connected genes are likely to be more pleiotropic (*Josephs et al., 2017*; *Proulx et al., 2005*; *He and Zhang, 2006*; *Hämälä et al., 2020a*). We used published information about the transcriptional regulatory network of *Drosophila*, which was inferred from several sources, including genome-wide chromatin immunoprecipitation, conserved transcription factor binding motifs, gene expression profiles across different development stages, and chromatin modification profiles among several cell types (*Marbach et al., 2012*). The connectivity for each gene was estimated as the sum of adjacencies between the focal gene and other genes in the network.

The significant positive correlation (rho = 0.54) between 1-$\tau$ and network connectivity suggests that both estimates capture a similar, but not identical, signature of the pleiotropic properties among the genes (*Appendix 1—figure 1*).

We note that approximating pleiotropic effects in *D. simulans* by using data-rich estimates from the close relative *Drosophila melanogaster* could potentially compromise the interpretability of our results. However, comparative analyses of tissue-specific gene expression have shown an extremely high consistency across species in animals (*Mao et al., 2024*) and plants (*Davidson et al., 2012*;

*Julca et al., 2021*). Hence, these data suggest that at least for tissue specificity, one of the pleiotropy measures used, estimates from one species can be safely used for close relatives.

## Experimental evolution and common garden experiment

The procedures of the evolution and common garden experiments were described in *Barghi et al., 2019*; *Hsu et al., 2020*; *Jakšić et al., 2020*. Briefly, 202 isofemale lines collected from Florida, USA were used to constitute 10 outbred *Drosophila simulans* populations, which have been exposed for more than 100 generations to a laboratory experiment at 28/18 °C with 12 hr light/12 hr dark photoperiod. The census population size of each replicate population is 1000–1250 adult flies.

Each founder isofemale line was maintained at a small population size (typically less than 50 individuals) at 18 °C on standard laboratory food. Adaptation to the laboratory environment with the residual heterogeneity or de novo mutation is unlikely due to the small effective population size during the maintenance of each line (*Barghi et al., 2019*). This is supported by the observation that 50 generations of maintenance as isofemale lines does not result in significant genetic differentiation from the ancestral population (*Nouhaud et al., 2016*). Furthermore, mutations that occur in the laboratory are mostly recessive (*Charlesworth and Charlesworth, 2010*), and the effects are likely to be masked because, after two generations of random mating during common garden maintenance, most individuals will be heterozygous for isofemale line-specific variants.

The collection of samples for RNA-Seq was preceded by two generations of common garden experiments (CGE). As described previously (*Barghi et al., 2019*; *Hsu et al., 2020*; *Jakšić et al., 2020*), at generation 103 of the evolution experiment, five biological replicates of reconstituted ancestral populations were generated by pooling five mated females from 184 founder isofemale lines (*Nouhaud et al., 2018*) and reared together with 10 independently evolved populations for two generations with controlled egg density (400 eggs/bottle) at the same temperature regime as in the evolution experiment. Three biological replicates were included for each independently evolved population (each biological replicate was cultivated separately for the two generations of the common garden). From each biological replicate, a sample of 50 5-d-old adult males was collected (*Figure 1a*).

## Quantifying the evolutionary parallelism in gene expression across ten replicate populations

The raw RNA-Seq count table of five replicates of the reconstituted ancestral population and 10 independently evolved populations, each with three biological samples, was taken from *Jakšić et al., 2020*. The raw read counts of each gene were normalized with the TMM method implemented in edgeR (*Robinson et al., 2010*). Only genes with at least 0.1 normalized counts per million base (CPM) across all samples were considered for further analysis. Because we are interested in the evolutionary response in each independently evolved population, we contrasted the three biological samples from each evolved population to the five biological samples from the ancestral population. The differential expression (DE) analysis was done separately in each of the ten evolved populations. For DE analysis, we utilized negative binomial generalized linear modeling implemented in edgeR to fit the expression to the model $Y = E + \varepsilon$, in which $Y$ stands for gene expression, $E$ is the effect of evolution and $\varepsilon$ is the random error. Likelihood ratio tests were performed to test the effect of evolution. p-values were adjusted using Benjamini-Hochberg false discovery rate (FDR) correction (*Benjamini and Hochberg, 1995*). After we identified DE genes in each evolved population, we constructed a replicate-frequency spectrum (RFS; *Barghi et al., 2019*). We considered genes with significant changes in the same direction in at least three evolved populations as putatively adaptive genes.

To quantify the degree of parallel gene expression evolution, we calculated the expression changes in response to evolution (log$_2$FC; fold change in expression between ancestral and evolved sample; see below) of each putatively adaptive gene (n=1775) in each evolved sample.

$$\log_2 FC = \log\left(\frac{y_{evo}}{\overline{y_{anc}}}\right),$$ (1)

where $y_{evo}$ is the expression value (CPM) per gene per evolved sample, and $\overline{y_{anc}}$ is the overall mean expression value (CPM) per gene across five ancestral samples.

We calculated the ratio of between-evolution population variation (MS$_{pop}$) and residual (MS$_e$) among the expression changes (log$_2$FC), which is denoted as F.

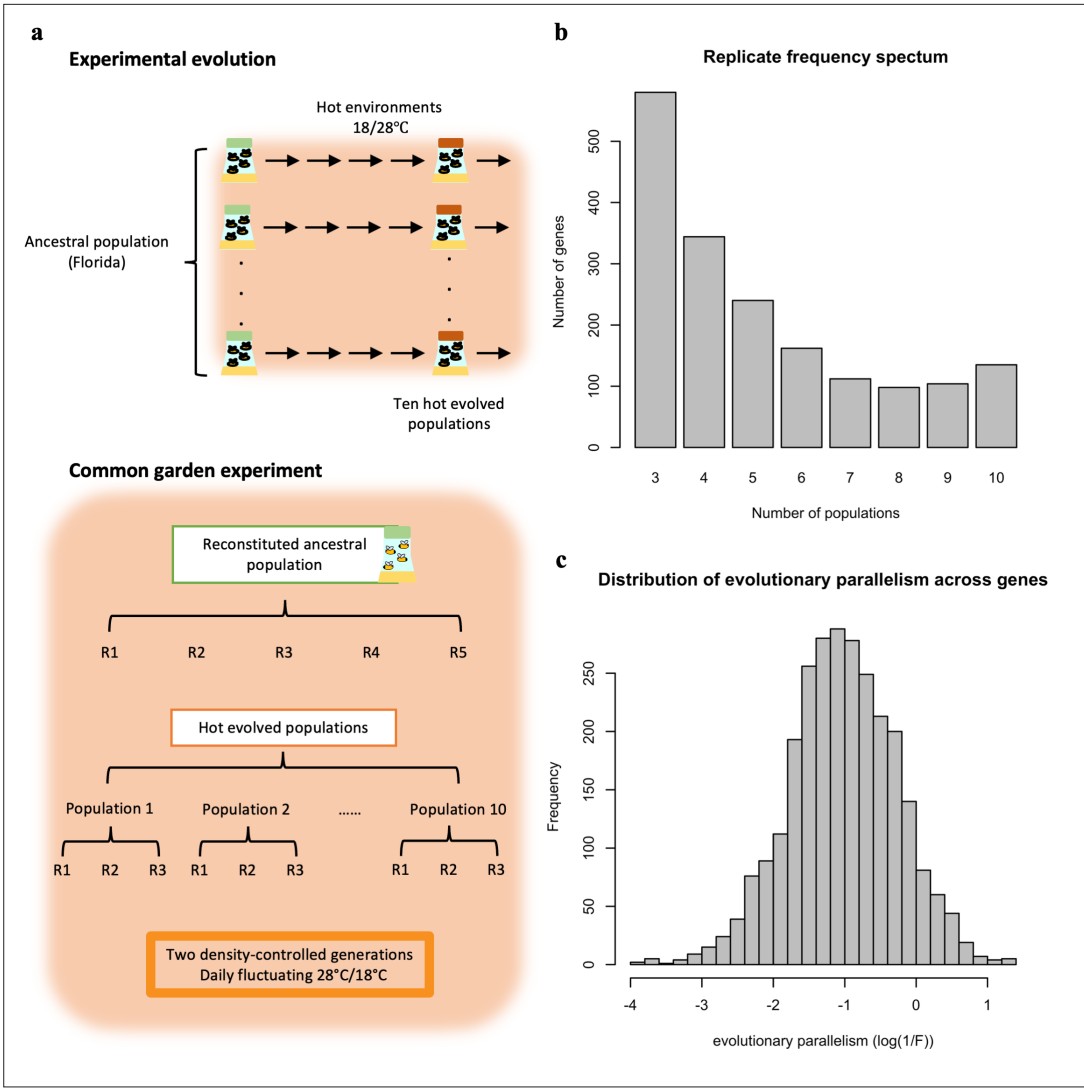

**Figure 1.** Overview of the experimental procedures (**a**) and the variation in parallel gene expression evolution across genes (**b,c**). (**a**) Experimental Evolution: ten replicated populations seeded from one common founder population have been evolving for >100 generations in a hot laboratory environment at 18 and 28°C. Common Garden Experiment: at the 103rd generation of the evolution experiment, the ten evolved populations (each with three biological replicates) were maintained together with the reconstituted ancestral population (with five biological replicates) for two generations in the same environment as in the evolution experiment. After two generations in the common garden, 50 males from each biological replicates were pooled and subjected to RNA-Seq. (**b**) Replicate frequency spectrum. Number of populations (x-axis) in which a given gene experienced a significant change in gene expression. The y-axis indicates the number of genes in each category. Most of the genes experienced a significant change in gene expression in a few evolved populations while much fewer genes were significant in all 10 populations. This pattern suggests that the parallelism in gene expression evolution differs across genes. (**c**) The distribution of gene expression evolution parallelism (log(1 /F)) across genes. Larger log(1 /F) values indicate more parallel evolution of a gene. The exhibit variation suggests that genes varied in their parallelism of its expression evolution.

$$F = \frac{MS_{pop}}{MS_e} \tag{2}$$

$$MS_{pop} = \frac{\sum_{i=1}^{10} (\bar{X}_i - \bar{X})^2}{10 - 1} \tag{3}$$

$$MS_e = \frac{\sum_{i=1}^{10} \sum_{j=1}^{3} (X_{ij} - \bar{X}_i)^2}{30 - 10}$$

Where $X_{ij}$ represent the log$_2$FC value for the j$^{th}$ biological sample in the i$^{th}$ evolution population. i=1, 2, …,10, j=1, 2, 3.

As F reflects the heterogeneity of the evolutionary response for a gene across 10 evolutionary populations, we used the reciprocal of F (i.e. 1 /F) to quantify the degree of parallelism in gene expression evolution.

### Estimating gene expression variation in an outbred ancestral population

The variation in expression of each gene within an outbred ancestral population was estimated from 20 adult males sampled from an outbred ancestral population (*Lai and Schlötterer, 2022*). We distinguished biological variation and measurement error of each gene using the statistical method implemented in edgeR *Robinson et al., 2010*, where the biological variance across individual (biological coefficient of variation; BCV; defined in *Robinson et al., 2010*) of each gene can be estimated (tagwised dispersion). BCV$^2$ was used to represent the expression variance relative to the mean of each gene within the population. To illustrate that the gene expression variation is not explained by random measurement error, we took the data from two ancestral population replicates, which are considered to be genetically identical, and calculated BCV$^2$ of expression levels across individuals for each population replicates separately. We found that both mean and variance in expression for each gene were similar in two ancestral population replicates (mean: rho >0.99; variance: rho = 0.8, p-value <2.2e-16), suggesting that sequencing noise does not mask the biological variability.

### Causal analysis

To infer the causal relationship between three factors: pleiotropy (Pl), ancestral variation (A), and parallelism (Pa), we applied causal analysis. The causal analysis is built upon a previously published statistical framework (*Schadt et al., 2005*). We consider five possible causal relationships between pleiotropic effects, ancestral variation, and parallelism (Figure 5). Our goal is to determine which of the five models is best supported by the data. The selection is based on Bayesian information criteria (BIC) which are calculated from the maximum likelihood of the respective models. The likelihood of each model is represented as:

$$\mathcal{L}\left(\text{Model I}\right) = p\left(Pa|A\right) p\left(A|Pl\right) p\left(Pl\right) \tag{5}$$

$$\mathcal{L}\left(\text{Model II}\right) = p\left(Pa|Pl\right) p\left(A|Pl\right) p\left(Pl\right) \tag{6}$$

$$\mathcal{L}\left(\text{Model III}\right) = p\left(Pa|A, Pl\right) p\left(A|Pl\right) p\left(Pl\right) \tag{7}$$

$$\mathcal{L}\left(\text{Model IV}\right) = p\left(Pa|A, Pl\right) p\left(A\right) p\left(Pl\right) \tag{8}$$

$$\mathcal{L}\left(\text{Model V}\right) = p\left(Pa\right) p\left(A\right) p\left(Pl\right) \tag{9}$$

p(Pl) is the probability distribution of the pleiotropic effect. To utilize the normal probability density function in the derivation of the likelihoods for each joint probability distribution, we transformed the random variables Pa and A to be normally distributed (natural log transformation on 1 /F and BCV$^2$). The exact forms of these likelihoods are given in the Appendix.

For each model, we maximized the likelihood and estimated the parameters using standard maximum likelihood methods. Based on the maximum likelihood, we computed the BICs for each model; the model with the smallest BIC is the one best supported by the data.

BIC is calculated as:

$$\text{BIC} = -2 \ln\left(\mathcal{L}_i\right) + k_i \ln\left(n\right) \tag{10}$$

Where $L_i$ is the likelihood for model I-V, $k_i$ is the corresponding number of free parameters and n is the total number of putatively selected genes. The analysis was performed independently on two measures of pleiotropy (tissue specificity and network connectivity).

## Estimating the sizes of direct and indirect pleiotropic effects on parallelism

We used path analysis to estimate the effect sizes of direct and indirect pleiotropic effects on parallelism (*Wright, 1920*). We standardized all three transformed variables (Parallelism (Pa), Pleiotropy (Pl), and Ancestral variation (A); see Appendix) by subtracting the mean and dividing by the standard deviation for the following analysis.

First, we fit the regression model for all three variables across all putatively adaptive genes:

$$y_{Pa} = \beta_1 x_A + \beta_2 x_{Pl} + \varepsilon \tag{11}$$

where $y_{Pa}$ stands for parallelism, $x_A$ is ancestral variation, $x_{Pl}$ is pleiotropy measure and $\varepsilon$ is a random error. $\beta_1$ and $\beta_2$ are the regression coefficients corresponding to $x_A$ and $x_{Pl}$, respectively.

Second, we fit another regression model for ancestral variation and pleiotropy measure across all putatively adaptive genes:

$$y_A = \beta_3 x_{Pl} + \varepsilon \tag{12}$$

where $y_A$ stands for ancestral variation, $x_{Pl}$ is pleiotropy measure and $\varepsilon$ is a random error. $\beta_3$ is the regression coefficient corresponding to $x_{pl}$.

The sizes of the direct effect of pleiotropy on parallelism will be $\beta_2$. The size of the indirect effect of pleiotropy on parallelism via ancestral variation will then be the product of $\beta_1 \times \beta_3$.

The analysis was performed independently on two different measures of pleiotropy.

## Computer simulations

In our interpretation of the indirect effect of pleiotropy on parallelism, we assumed that ancestral variation in gene expression is negatively correlated with its parallel evolution. Because this has not been previously demonstrated, we used computer simulations to illustrate how the level of standing genetic variation impacts the parallelism of adaptive responses after a shift in trait optimum. For simplicity, we considered four redundant genes equally contributing to a fitness-associated trait. The phenotype (expression level) of each gene is controlled by a different number of genetic loci (5, 15, 30, and 50) with equal effect. The underlying assumption is that these four genes are redundant in terms of their fitness effect during the experimental evolution, but they exhibit different levels of pleiotropy and thus have historically experienced different levels of purifying selection, leaving different levels of genetic variation at the start of the experiment. Using mimicrEE2 (*Vlachos and Kofler, 2018*), we evolved 10 replicate populations (N=300) derived from the same set of 189 natural *Drosophila* haplotypes (*Barghi et al., 2019*) in the same selection regime with a shift in trait optimum of one standard deviation relative to the ancestral phenotypic distribution. After 100 generations, we determined the evolutionary parallelism of the four genes using the same approach as for the empirical data. Each simulation was repeated 100 times and one out of four genes was randomly picked from each simulation run to generate Figure 4 (i.e. we estimated variance in the ancestral population and parallelism for the evolved replicate populations).

## Results

### The parallelism of expression evolution differs among genes

We investigated the degree of parallelism in gene expression evolution, using an RNA-Seq dataset from an experimental evolution study in which 10 replicated populations with the same genetic variation were exposed to the same environment for more than 100 generations (*Hsu et al., 2021*; *Jakšić et al., 2020*; *Lai and Schlötterer, 2022*; *Figure 1a*). While previous studies have focused on the shared parallel response across replicated populations (*Hsu et al., 2021*; *Jakšić et al., 2020*; *Lai and Schlötterer, 2022*), we re-analyzed the RNA-Seq data to quantify the degree of evolutionary parallelism across different genes. We identified 1775 putatively selected genes with a significant change in gene expression in the same direction in response to the novel environment in at least three

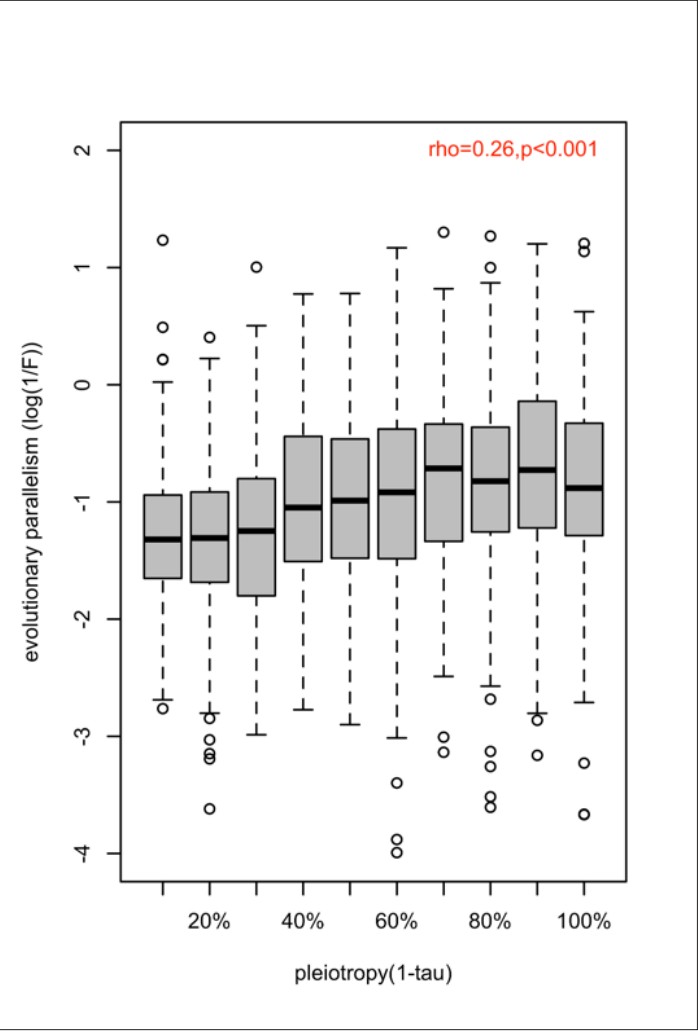

**Figure 2.** Association between the magnitude of evolution parallelism and the strength of pleiotropy. The distribution of evolution parallelism (log(1 /F)) of putatively adaptive genes (n=1775) is shown in boxplots binned by their strength of pleiotropy (1-$\tau$). Overall, the strength of pleiotropy was positively correlated with evolution parallelism (spearman correlation coefficient; $\rho$=0.26; p-value <2.2e-16).

populations. This conditioning on multiple replicate populations with a significant change in the same direction reduces false positives due to drift (*Rennison and Peichel, 2022*; *Bohutínská et al., 2021*). Relaxing the threshold of responding populations provides qualitatively similar results.

*Barghi et al., 2019* introduced the replicate frequency spectrum to describe the parallel genomic selection response in replicate populations. The replicate frequency spectrum of DE genes shows considerable heterogeneity among populations, with many genes displaying significant gene expression changes only in a few populations (*Figure 1b*). The replicate frequency spectrum is strongly influenced by the significance threshold used and provides an incomplete quantitative measure of parallelism. To obtain a more quantitative measure of parallelism, we evaluated the between-population variation in the evolutionary response ($\log_2$FC) of each putatively adaptive gene (n=1775) while controlling for measurement error (F). The parallelism of evolution for each gene is then quantified as 1 /F (*Figure 1b*; see Materials and Methods). A higher value of 1 /F indicates a more parallel evolutionary response among replicated populations (*Appendix 1—figure 2*). Importantly, the measure of parallelism varies between genes (*Figure 1c*) and is positively correlated with the number of populations in which a significant change in gene expression is observed for each gene (*Appendix 1—figure 3*).

## Pleiotropy is positively associated with the degree of parallelism in gene expression evolution

Pleiotropy is one of the factors that influence the degree of parallel evolution (*Bolnick et al., 2018*; *Conte et al., 2015*; *Rennison and Peichel, 2022*; *Stern, 2013*). Therefore, we tested whether and how the pleiotropic effect of a gene influences the degree of parallelism of its expression evolution in a new environment. We correlated two different measures of pleiotropy (1-$\tau$ (tissue specificity) and network connectivity; see materials and methods) of each putatively adaptive gene (n=1775) with the parallelism of its evolutionary responses (1 /F). We observed a significant positive correlation between the parallelism and the degree of pleiotropy, (rho = 0.26, p-value <2.2e-16 for 1-$\tau$ and rho = 0.11, p-value <5.6e-07 for network connectivity) (*Figure 2* and *Appendix 1—figure 4a*). This suggests that pleiotropy may enhance parallel changes in gene expression during evolution.

## Pleiotropy constrains the ancestral (natural) variation in gene expression

The negative effect of pleiotropy on genetic variation over longer time scales, mostly between species, is well documented (*Fraser et al., 2002*; *Hahn and Kern, 2005*; *Masalia et al., 2017*; *Josephs et al., 2017*), but it is not yet clear whether the same effect is seen for adaptation from standing genetic variation. We reconstituted the ancestral population from 189 isofemale lines (*Lai and Schlötterer, 2022*) and estimated the expression variance (BCV$^2$; see materials and methods) of each putatively adaptive gene (n=1775) in two replicate sets of 20 individuals. Since the variance in gene expression

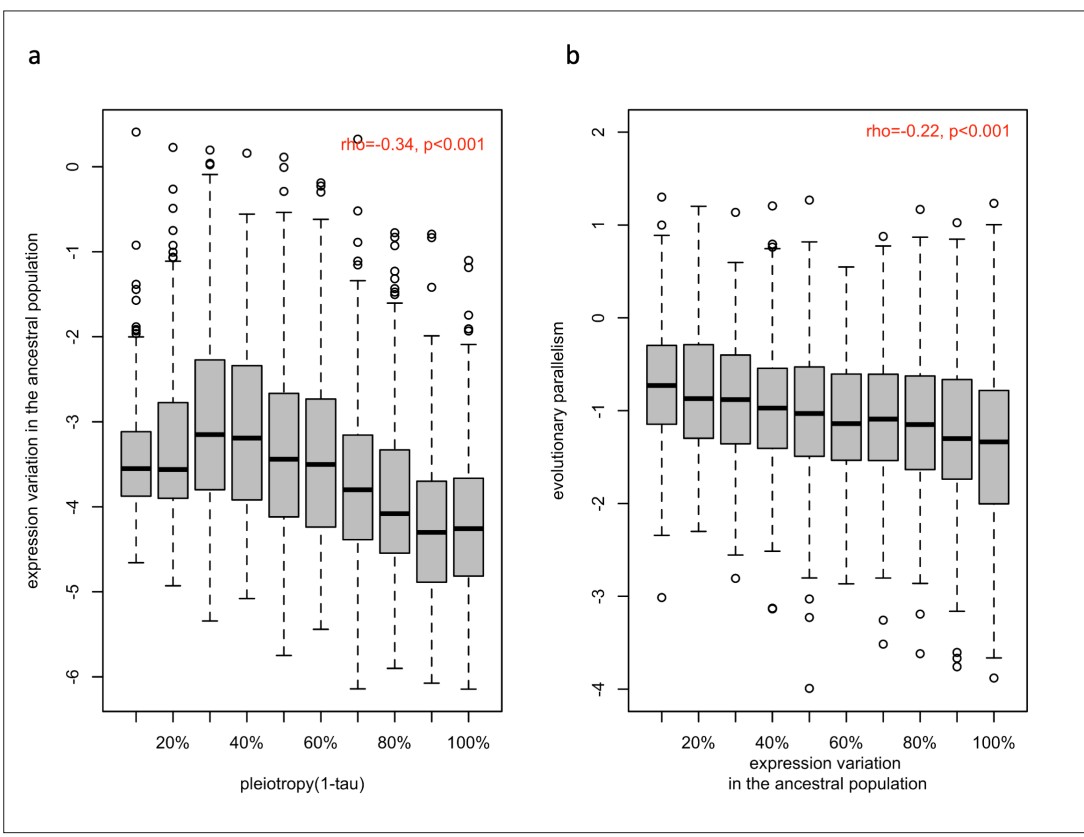

**Figure 3.** Association between ancestral variation in gene expression with the strength of pleiotropy (1-$\tau$) (**a**) and with the evolution parallelism (**b**). (**a**) The distribution of gene expression variation in the ancestral population (log (BCV$^2$)) of putatively evolved genes (n=1,775) is shown in boxplots binned by their strength of pleiotropy (1-$\tau$). Overall, the strength of pleiotropy was negatively correlated with ancestral variation in gene expression (spearman correlation coefficient; $\rho=-0.34$, p-value <2.2e-16). (**b**) The distribution of evolutionary parallelism (log(1 /F)) of putatively adaptive genes (n=1775) is shown in boxplots binned by their strength of ancestral variation. The strength of parallelism is negatively correlated with the ancestral variation in gene expression (spearman correlation coefficient; $\rho=-0.22$, p-value <2.2e-16).

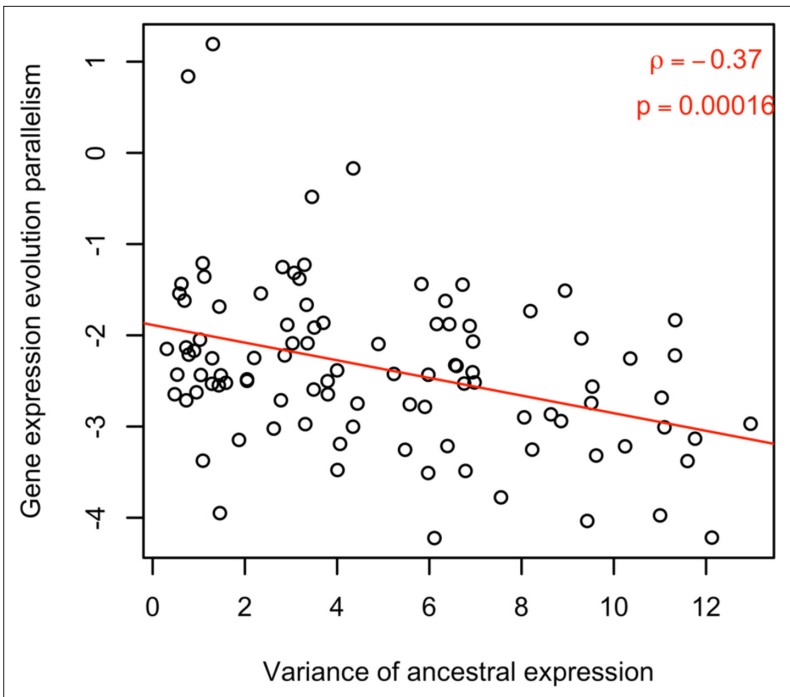

**Figure 4.** Computer simulations illustrate the negative effect of ancestral variation on parallel evolution. Computer simulations assume a fitness-related trait determined by the expression of four genes. The expression of these four genes is determined by a different number of genetic loci, reflecting the influence of pleiotropy on ancestral genetic variation. A shift in trait optimum was used to illustrate the connection between ancestral variation in expression (x-axis) and parallelism of expression change across ten evolved populations (y-axis). The negative correlation between them ($\rho$ = –0.26, p-value <1.77e-07) suggests that the genes with less ancestral variation are resulting in more parallel responses.

is positively correlated with genetic variation in cis-regulatory regions (*Hämälä et al., 2020b*) and the heritability of gene expression is high (*Ayroles et al., 2009*), we assumed that the observed variation in gene expression has a genetic basis, rather than being noise (see discussion). We observed a highly significant negative correlation between the level of pleiotropy and the magnitude of ancestral variation in gene expression (*Figure 3a* and *Appendix 1—figure 4b*; rho = −0.34, p-value <2.2e-16 for 1-$\tau$ and rho = −0.37, p-value <2.2e-16 for network connectivity). To exclude that this correlation was driven by our filtering for significant expression changes in at least three populations, we repeated these analyses with all genes (n=9,882) and also observed a significant correlation (rho = −0.45, p-value <2.2e-16 for 1-$\tau$ and rho = −0.38, p-value <2.2e-16 for network connectivity, data not shown). Hence, we suggest that the negative relationship between pleiotropy and polymorphism/ diversity (*Lemos et al., 2004*; *Mähler et al., 2017*) (i.e. reduced variants, lower frequencies, and/or lower allelic effects) can be extended to gene expression.

## Evolutionary parallelism is negatively associated with ancestral variation

For a better understanding of the factors determining parallel evolution, we examined the influence of ancestral variation in gene expression on the parallel evolution of gene expression. We observed a significant negative correlation between ancestral variation and parallelism of the evolutionary response (rho = −0.22, p-value <2.2e-16; *Figure 3b*). We suggest that this pattern could be explained by genetic redundancy (*Barghi et al., 2019*; *Láruson et al., 2020*), which describes the phenomenon that a polygenic trait can shift its phenotypic value through different combinations of contributing loci: When the diversity of the contributing loci is reduced (either through fewer loci, lower ancestral frequencies or smaller variation in effect size), fewer combinations of the contributing loci are available for adaptation, resulting in more parallel adaptive responses. To confirm this, we performed computer simulations of replicated polygenic adaptation and found that lower ancestral phenotypic

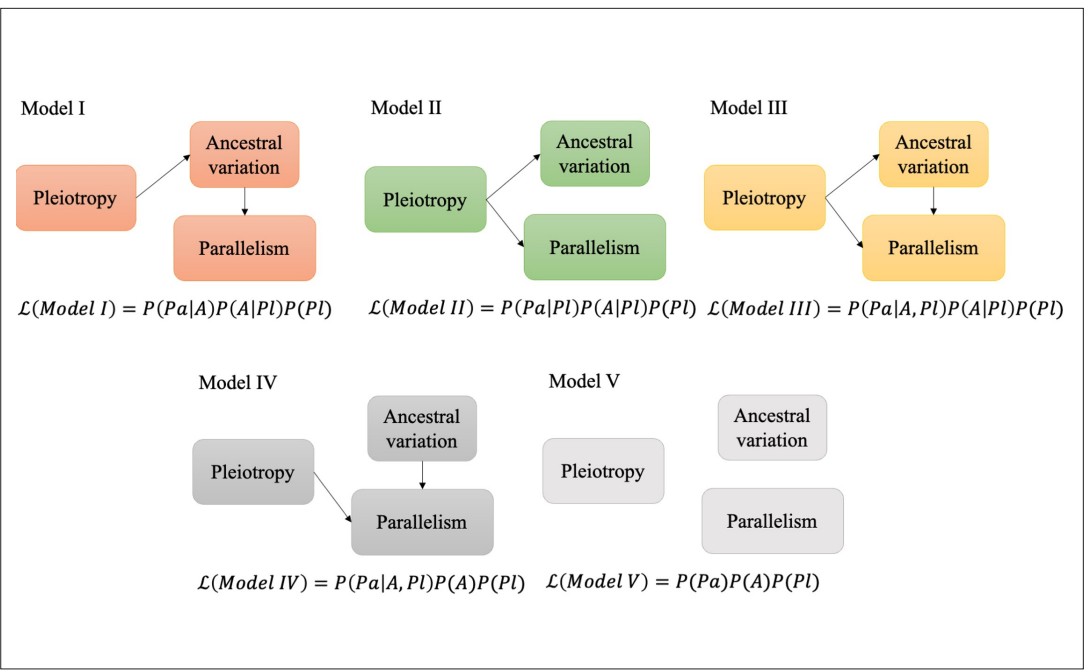

**Figure 5.** Schematic figure illustrating the causal models evaluated. Five possible relationships between pleiotropic effects, ancestral variation, and evolutionary parallelism. $\mathcal{L}$ denotes the likelihood of each model given the data. P is the probability or conditional probability of the measurements for each gene; $Pa$ is the parallelism level and $A$ is the level of ancestral variation in gene expression. $Pl$ is the pleiotropic effects. See materials and methods for a more detailed description.

(i. e. expression) variation leads to more parallel trait evolution (*Figure 4*). Hence, that natural variation in the ancestral population could be one of the factor determining the parallel adaptive response in replicate populations. Although we focused on the phenotypic response to match gene expression, it is interesting to note that a similar result was obtained for parallelism at the genetic level with different levels of standing genetic variation (*Thompson et al., 2019*). If pleiotropy reduces variation, the positive correlation between pleiotropy and parallelism may be an indirect (correlative) effect, rather than a direct consequence of pleiotropy.

## Testing causality

Our previous analyses indicated that parallelism is affected by pleiotropy and ancestral variation. Because pleiotropy is negatively correlated with ancestral variation for both pleiotropy measures (*Figure 3a* and *Appendix 1—figure 4b*), it is not clear whether pleiotropy affects parallelism directly or indirectly through the negative correlation with ancestral variation. We used causal analysis, which is based on a Bayesian statistical framework (*Schadt et al., 2005*), to disentangle the causal relationship between the three different parameters. Since the directionality of pleiotropic effects on ancestral variation has been previously demonstrated, and ancestral variation is unlikely to influence

**Table 1.** Bayesian Information Criteria (BIC) value for Model I-V.
The model with the smallest BIC is the one best support by the data.

| Model | BIC value (Network connectivity) | BIC value (Tissue specificity) |
|---|---|---|
| I | 9258.395 | 8958.125 |
| II | 9316.852 | 8972.626 |
| III | 9264.769 | 8939.342 |
| IV | 9490.017 | 9162.090 |
| V | 9550.309 | 9247.540 |

**Table 2.** The size of direct and indirect pleiotropic effects on the evolutionary parallelism.

| | Direct effect | Indirect effect |
|---|---|---|
| 1-$\tau$ | 0.127 | 0.055 (−0.181, −0.332) * |
| Network connectivity | 0.026 | 0.068 (−0.210, −0.337) * |

*Coefficient estimates (β2,β3) whose product provides the indirect effect.

the pleiotropy of a gene, we considered five possible causal relationships between pleiotropic effects, ancestral variation, and parallelism (*Figure 5*). In the first model, parallelism is determined by ancestral variation, which is shaped by pleiotropic, but pleiotropy has no direct effect on parallelism. In model II, the pleiotropic effect affects ancestral variation and parallelism independently, but ancestral variation has no causal effect on parallelism. In model III, pleiotropy affects both ancestral variation and parallelism, and ancestral variation also determines parallelism. In model IV, parallelism is determined by ancestral variation and pleiotropy, but pleiotropic effects are assumed to have no effect on the ancestral variation. This serves as a null model to confirm the indirect effect of pleiotropy on parallelism via ancestral variation. We also add a global null model with no correlation between these three factors (model V, *Figure 5*). The analysis was performed independently for two measures of pleiotropy (tissue specificity and network connectivity).

For tissue specificity, our data were better explained by model III (i.e. model III has the lowest BIC, *Table 1*), suggesting that both direct and indirect effects of pleiotropy determine the degree of parallelism of adaptive gene expression evolution. On the other hand, when pleiotropy was estimated by network connectivity, model I had a slightly lower BIC than model III (*Table 1*). Thus, our causal analysis does not provide evidence for a direct effect of network connectivity. We further quantified the strength of the direct and indirect effects of pleiotropy on parallelism using a path analysis (see materials and methods). The path analysis confirmed the significant direct effect of tissue specificity, but network connectivity has no statistically significant direct effect on the parallelism of gene expression evolution (*Table 2*, *Appendix 1—table 1*). Although the relevance of the direct effects of pleiotropy on parallel gene expression evolution differs for the two pleiotropy estimates, the similarity of the indirect effect sizes for both pleiotropy measures is striking (*Table 2*). Taken together, these results suggest that pleiotropy may enhance the parallel evolutionary response of gene expression directly and indirectly through its influence on ancestral variation.

## Discussion

This study examined the influence of pleiotropy on the parallelism of the adaptive response from standing genetic variation. By examining the role of natural variation, which is important for parallelism of short-term evolution, we extended the scope of previous studies. Because differences in genetic background and selection strength among replicated populations cannot be controlled in natural populations and may affect the degree of parallelism, we relied on experimental evolution for our study. We took advantage of ten replicated populations that have evolved from the same genetic background in the same hot environment for more than 100 generations (*Barghi et al., 2019*).

The adaptive response during adaptation to the new hot temperature regime was measured by changes in gene expression. This approach has the advantage that pleiotropic effects and their associated fitness function, which are typically not known, are already included in the adaptive response. Furthermore, we were able to rely on two well-established measures of pleiotropy to study the interplay between parallelism and natural variation.

A key finding of our study is that adaptation from standing genetic variation results in complex interplays between pleiotropy, ancestral variation, and parallelism. Of particular interest is the observed negative correlation between pleiotropy and ancestral variation in gene expression, which confirms previous observations (*Lemos et al., 2004*; *Mähler et al., 2017*). This suggests that the primary effect of pleiotropy is not to maintain variation. Rather, stronger purifying selection removes pleiotropic variants from the population, resulting in a reduced standing genetic variation. The reduced variation in more pleiotropic genes could reflect historical hard sweeps or purifying selection. A similar pattern

is seen over longer evolutionary timescales: less sequence divergence in pleiotropic genes (*Fraser et al., 2002*; *Hahn and Kern, 2005*; *Masalia et al., 2017*; *Josephs et al., 2017*).

An important difference between adaptation from de novo mutations and standing genetic variation concerns the implications for parallel evolution. While pleiotropic genes are less likely to contribute to parallel evolution from de novo mutations (*Bolnick et al., 2018*), our results show that adaptation from standing genetic variation favors parallel selection responses of pleiotropic genes (*Figure 2*, *Appendix 1—figure 4a*). It is important to note that many previous studies focused on sequence variation, whereas we used expression variation. While it is possible that these two types of variation behave differently, we do not believe that this is the case as sequence polymorphism and expression variation are correlated (*Lawniczak et al., 2008*). Furthermore, a recent study focusing on genomic changes (*Rennison and Peichel, 2022*) reached similar conclusions.

A very important assumption in our interpretation is that differences in gene expression variation in the ancestral population reflect regulatory sequence polymorphism rather than noise. The high correlation in the variance of gene expression between two replicate groups of individuals indicates that our measures do not reflect technical noise but have a biological basis. However, stochastic fluctuations in gene expression have also been observed among cells in single-celled and multicellular organisms and the variance in gene expression differs among genes (*Swain et al., 2002*; *Elowitz et al., 2002*; *Blake et al., 2003*). In this case, the differences in gene expression variance are biological in nature, but not associated with sequence variation. We do not believe that variation in the control of gene expression levels among genes can explain our pattern because *Drosophila* is a multicellular organism and we used whole body RNA-Seq data for our analysis. Therefore, even if expression is controlled to different extents among genes, by averaging across multiple cells, this does not affect our results. The only scenario we can imagine in which gene expression noise could propagate across the many cells of a multicellular organism is that during early development, differences in gene expression of a key regulator could affect downstream gene expression throughout the entire organism. However, we do not think this is a likely scenario, especially in the light of previous results showing that genetic variation in cis-regulatory regions correlates with expression variation (*Hämälä et al., 2020b*). Furthermore, whole-body gene expression in *Drosophila* is highly heritable, suggesting that gene expression is not stochastic (*Ayroles et al., 2009*). Nevertheless, we would like to point out that our observations are valid irrespective of whether the correlation between parallelism and expression variation is the result of genetic polymorphism or genetic control of expression noise.

Based on the significant correlation results, the question arose as to whether the positive correlation between pleiotropy and parallelism, is causal (direct effect) or merely correlational (indirect effect). Regardless of the measure of pleiotropy, our causal analysis showed a consistent indirect effect of pleiotropy on parallelism (*Tables 1 and 2*). The estimated indirect effects were very similar for both measures of pleiotropy (*Table 2*). This indirect effect of pleiotropy could be explained by the reduced variation in the expression of more pleiotropic genes in the ancestral population leading to a more parallel selection response of gene expression. For the direct effect, however, the results differed between the two measures of pleiotropy. Only tissue specificity had a significant direct effect, which was even larger than the indirect effect (*Table 2*). No significant direct effect was found for network connectivity. The discrepancy between the two measures of pleiotropy is particularly interesting given their significant correlation (*Appendix 1—figure 1*). This suggests that both measures capture aspects of pleiotropy that differ in their biological implications. The direct effects of pleiotropy, most likely arise from the positively correlated fitness effects of non-focal traits (synergistic pleiotropy). The synergistic pleiotropic effects result in a stronger fitness advantage, which translates into a more parallel selection response - as previously shown (*MacPherson and Nuismer, 2017*). Given that temperature is one of the most important abiotic factors, it is expected that positive genetic correlations between temperature-adaptive traits have been selected over long evolutionary timescales, probably prior to the speciation of *D. simulans*. Thus, it is conceivable that positive pleiotropy for temperature adaptation-related traits is common and thus determines the positive contribution of pleiotropy to the parallel evolution observed here.

We focused our analysis on genes that exhibited evidence of adaptive responses, defined as significant changes in the same direction across at least three populations. This approach allowed us to investigate the extent of parallel adaptive responses and how pleiotropy contributes to variation in parallelism among genes. Interestingly, when comparing pleiotropy between DE genes (those showing

adaptive responses) and non-DE genes (those that did not change their response during evolution), we observed significantly higher pleiotropy in the non-DE gene group (*Appendix 1—figure 5*). This finding suggests that extremely high levels of pleiotropy may constrain evolutionary responses. Overall, the pattern of increased pleiotropy associated with evolutionary parallelism supports the idea that low to intermediate levels of pleiotropy are more favorable for adaptation (*Rennison and Peichel, 2022*; *Frachon et al., 2017*).

In a concluding remark, we would like to emphasize that although pleiotropy and ancestral variation were found to be significantly associated with the degree of parallelism of gene expression evolution, it is likely that additional factors also contribute. For example, diminishing return epistasis, population size (drift), and the contribution of gene expression to the selected traits may also affect the parallelism of gene evolution. Further investigation of other potential factors will continue to advance our understanding of parallelism and the predictability of evolution.

## Acknowledgements

We thank Viola Nolte for preparing all RNA-Seq libraries and supervising the maintenance of the evolution experiment. We thank Dagný Ásta Rúnarsdóttir for the suggestions on the early version of the manuscript and all members of the Institut für Populationsgenetik for discussion. Ana Marija Jakšić, Neda Barghi, François Mallard, and Kathrin Otte performed the common garden experiment. Illumina sequencing was done at the VBCF NGS Unit https://www.viennabiocenter.org/vbcf/. This research was funded in whole or in part by the Austrian Science Fund (FWF) [W1225, P32935] and the European Research Council (ERC) [ArchAdapt]. For open access purposes, the author has applied a CC BY public copyright license to any author-accepted manuscript version arising from this submission.

## Additional information

### Funding

| Funder | Grant reference number | Author |
| --- | --- | --- |
| Austrian Science Fund | 10.55776/W1225 | Christian Schlötterer |
| European Research Council | Archadapt | Christian Schlötterer |
| Austrian Science Fund | 10.55776/P32935 | Christian Schlötterer |

The funders had no role in study design, data collection and interpretation, or the decision to submit the work for publication.

### Author contributions

Wei-Yun Lai, Conceptualization, Formal analysis, Writing – original draft, Writing – review and editing; Sheng-Kai Hsu, Conceptualization, Formal analysis, Writing – review and editing; Andreas Futschik, Writing – review and editing, Statistical advice; Christian Schlötterer, Conceptualization, Supervision, Funding acquisition, Writing – original draft, Project administration, Writing – review and editing

### Author ORCIDs

Wei-Yun Lai https://orcid.org/0000-0002-5101-8695
Sheng-Kai Hsu https://orcid.org/0000-0002-6942-7163
Christian Schlötterer https://orcid.org/0000-0003-4710-6526

Reviewer #1 (Public review): https://doi.org/10.7554/eLife.102321.3.sa1
Reviewer #2 (Public review): https://doi.org/10.7554/eLife.102321.3.sa2
Reviewer #3 (Public review): https://doi.org/10.7554/eLife.102321.3.sa3
Author response https://doi.org/10.7554/eLife.102321.3.sa4

## Additional files

### Supplementary files
MDAR checklist

### Data availability
All sequencing data are available in European Nucleotide Archive (ENA) under the accessionnumber PRJEB35504, PRJEB35506 and PRJEB37011. Other raw data and scripts are available on GitHub (https://github.com/cloudweather34/pleiotropy, copy archived at *cloudweather34, 2025*).

The following previously published datasets were used:

| Author(s) | Year | Dataset title | Dataset URL | Database and Identifier |
|---|---|---|---|---|
| Jakšić AM, Karner J, Nolte V, Hsu SK, Barghi N, Mallard F, Otte KA, Svečnjak L, Senti KA, Schlötterer C | 2020 | Neuronal function and dopamine signaling evolve at high temperature in *Drosophila* | https://www.ebi.ac.uk/ena/browser/view/PRJEB35504 | EBI European Nucleotide Archive, PRJEB35504 |
| Hsu SK, Jakšić AM, Nolte V, Lirakis M, Kofler R, Barghi N, Versace E, Schlötterer C | 2020 | Rapid sex-specific adaptation to high temperature in *Drosophila* | https://www.ebi.ac.uk/ena/browser/view/PRJEB35506 | EBI European Nucleotide Archive, PRJEB35506 |
| Lai WY, Schlötterer C | 2020 | Evolution of phenotypic variance in response to a novel hot environment | https://www.ebi.ac.uk/ena/browser/view/PRJEB37011 | EBI European Nucleotide Archive, PRJEB37011 |

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

# Appendix

## Data transformation and standardization

As described in the main text, we quantified the evolutionary parallelism with 1 /F and the estimates were log-transformed to fit normal assumption.

$$Pa = \log\left(1/F\right) \tag{A1}$$

We measured the ancestral variation of gene expression with a log-transformed squared biological coefficient of variation based on the individual expression data of each gene for the ancestral population.

$$A = \log\left(BCV^2\right) \tag{A2}$$

The pleiotropy of a gene is approximated with the tissue-specificity of its expression and network connectivity. For each measure, 10 bins of equal sizes were derived based on deciles, and genes in different bins are considered to exhibit different extent of pleiotropy.

$$Pl = j, \text{when } \widehat{Pl}_{10(j-1)\%} < \widehat{Pl}_i \leq \widehat{Pl}_{10(j)\%} \tag{A3}$$

Where $j$=1, 2, 3…, 10; $Pl_i$ is 1-$\tau$ or connectivity of a gene; i=1, 2, 3…, n; n is the total number of genes.

We standardized all three transformed variables (Parallelism (Pa), Pleiotropy (Pl), Ancestral variation (A)) by subtracting the mean and dividing by the standard deviation for the causal and regression analysis.

## Exact forms of the likelihood for a causal model

To uncover the causal relationship between the genetic variables of interest (i.e. evolutionary parallelism (Pa), ancestral variation (A), and pleiotropy (Pl)), we represent the correlation structures in five different models considered in the main text (**Figure 4**) with Gaussian likelihoods and use Bayesian Information Criteria (BIC) to determine which model is best supported by the data. Similar approach has been taken previously to infer causality among genetic features of interest (**Schadt et al., 2005**).

The parameterization and likelihoods for each model over all putatively adaptive genes are given by:

$$\mathcal{L}\left(Model\,I\right) = \prod_{i=1}^{n}\sum_{j=1}^{k} P\left(Pl_j\right) l\left(\theta; A_i|Pl_j\right) l\left(\theta; Pa_i|A_i\right)$$

$$= \prod_{i=1}^{n}\sum_{j=1}^{k} P\left(Pl_j\right) \frac{1}{\sqrt{2\pi\sigma_A^2}} exp\left(-\frac{\left(A_i - \mu_{A_{Pl_j}}\right)^2}{2\sigma_A^2}\right) \tag{A4}$$

$$\frac{1}{\sqrt{2\pi\sigma_A^2\left(1-\rho^2\right)}} exp\left(-\frac{\left(Pa_i - \mu_{Pa} - \rho\frac{\sigma_A}{\sigma_{Pa}}\left(A_i - \mu_A\right)\right)^2}{2\sigma_{Pa}^2\left(1-\rho^2\right)}\right)$$

$$\mathcal{L}\left(Model\,II\right) = \prod_{i=1}^{n}\sum_{j=1}^{k} P\left(Pl_j\right) l\left(\theta; A_i|Pl_j\right) l\left(\theta; Pa_i|Pl_j\right)$$

$$= \prod_{i=1}^{n}\sum_{j=1}^{k} P\left(Pl_j\right) \frac{1}{\sqrt{2\pi\sigma_A^2}} exp\left(-\frac{\left(A_i - \mu_{A_{Pl_j}}\right)^2}{2\sigma_A^2}\right)$$

$$\frac{1}{\sqrt{2\pi\sigma_{Pa}^2}} exp\left(-\frac{\left(A_i - \mu_{Pa_{Pl_j}}\right)^2}{2\sigma_{Pa}^2}\right) \tag{A5}$$

$$\mathcal{L}\left(Model\,III\right) = \prod_{i=1}^{n}\sum_{j=1}^{k} P\left(Pl_j\right) l\left(\theta; A_i | Pl_j\right) l\left(\theta; Pa_i \mid A_i, Pl_j\right)$$

$$= \prod_{i=1}^{n}\sum_{j=1}^{k} P\left(Pl_j\right) \frac{1}{\sqrt{2\pi\sigma_A^2}} exp\left(-\frac{\left(A_i - \mu_{A_{Pl_j}}\right)^2}{2\sigma_A^2}\right)$$

$$\frac{1}{\sqrt{2\pi\sigma_{Pa}^2\left(1-\rho^2\right)}} exp\left(-\frac{\left(Pa_i - \mu_{Pa_{Pl_j}} - \rho\frac{\sigma_A}{\sigma_{Pa}}\left(A_i - \mu_A\right)\right)^2}{2\sigma_A^2\left(1-\rho^2\right)}\right) \tag{A6}$$

$$\mathcal{L}\left(Model\,IV\right) = \prod_{i=1}^{n}\sum_{j=1}^{k} P\left(Pl_j\right) l\left(\theta; A_i\right) l\left(\theta; Pa_i \mid A_i, Pl_j\right)$$

$$= \prod_{i=1}^{n}\sum_{j=1}^{k} P\left(Pl_j\right) \frac{1}{\sqrt{2\pi\sigma_A^2}} exp\left(-\frac{\left(A_i - \mu_A\right)^2}{2\sigma_A^2}\right)$$

$$\frac{1}{\sqrt{2\pi\sigma_{Pa}^2\left(1-\rho^2\right)}} exp\left(-\frac{\left(Pa_i - \mu_{Pa_{Pl_j}} - \rho\frac{\sigma_A}{\sigma_{Pa}}\left(A_i - \mu_A\right)\right)^2}{2\sigma_A^2\left(1-\rho^2\right)}\right) \tag{A7}$$

$$\mathcal{L}\left(Model\,V\right) = \prod_{i=1}^{n}\sum_{j=1}^{k} P\left(Pl_j\right) l\left(\theta; A_i\right) l\left(\theta; Pa_i\right)$$

$$= \prod_{i=1}^{n}\sum_{j=1}^{k} P\left(Pl_j\right) \frac{1}{\sqrt{2\pi\sigma_A^2}} exp\left(-\frac{\left(A_i - \mu_A\right)^2}{2\sigma_A^2}\right)$$

$$\frac{1}{\sqrt{2\pi\sigma_{Pa}^2}} exp\left(-\frac{\left(Pa_i - \mu_{Pa}\right)^2}{2\sigma_{Pa}^2}\right) \tag{A8}$$

Where i indicates each gene, j represents different pleiotropic levels, n is the total number of putatively selected genes and k equals to 10 in this study. For each likelihood model, the corresponding likelihood is maximized, and parameters are estimated using standard maximum likelihood methods.

The BICs are then computed for each model as follows:

$$BIC = -2ln\left(\mathcal{L}_i\right) + k_i ln\left(n\right) \tag{A9}$$

Where $L_i$ are the likelihood for model I-III, $k_i$ is the corresponding number of free parameters and n is a total number of putatively selected genes.

The model with the smallest BIC value is identified as the best-supported model.

## Regression models including the effects of two measures of pleiotropy on ancestral variation/parallelism

To examine whether two measures of pleiotropy explain ancestral variation/parallelism through common or independent features, we performed two additional regression analyses (*Equation A10; A11*). All variables were transformed and standardized (see Appendix).

First, to understand whether two pleiotropy measures explain the parallelism directly through common or independent features, we fit the regression model across all putatively adaptive genes as follows:

$$y_{Pa} = \beta_A x_A + \beta_{TS} x_{TS} + \beta_{NC} x_{NC} + \varepsilon \tag{A10}$$

Where $y_{Pa}$ stands for parallelism, $x_A$ is ancestral variation, $x_{TS}$ is 1-$\tau$, $x_{NC}$ is network connectivity and $\varepsilon$ is random error. $\beta_A$, $\beta_{TS}$ and $\beta_{NC}$ are the regression coefficients corresponding to $x_A$, $x_{TS}$ and $x_{NC}$, respectively. The regression coefficients for each corresponding variable are shown in **Appendix 1—table 1**.

Second, to understand whether two pleiotropy measures explain ancestral variation through the common or independent feature, we fit another regression model across all putatively adaptive genes as follows:

$$y_A = \beta_{TS}x_{TS} + \beta_{NC}x_{NC} + \varepsilon \tag{A11}$$

Where $y_A$ stands for ancestral variation, $x_{TS}$ is 1-$\tau$, $x_{NC}$ is network connectivity and $\varepsilon$ is random error. $\beta_{TS}$ and $\beta_{NC}$ are the regression coefficients corresponding to $x_{TS}$ and $x_{NC}$, respectively. The regression coefficients for each corresponding variable are shown in **Appendix 1—table 2**.

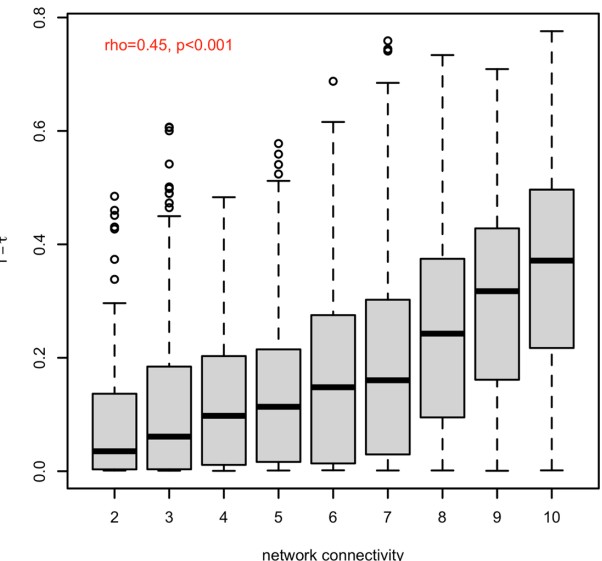

**Appendix 1—figure 1.** Positive correlation between two measures of pleiotropy. The substantial correlation (rho = 0.45, p-value <2.2e-16) between network connectivity (x-axis) and 1- $\tau$ (y-axis) suggests that both estimates are capturing similar information representing the pleiotropy.

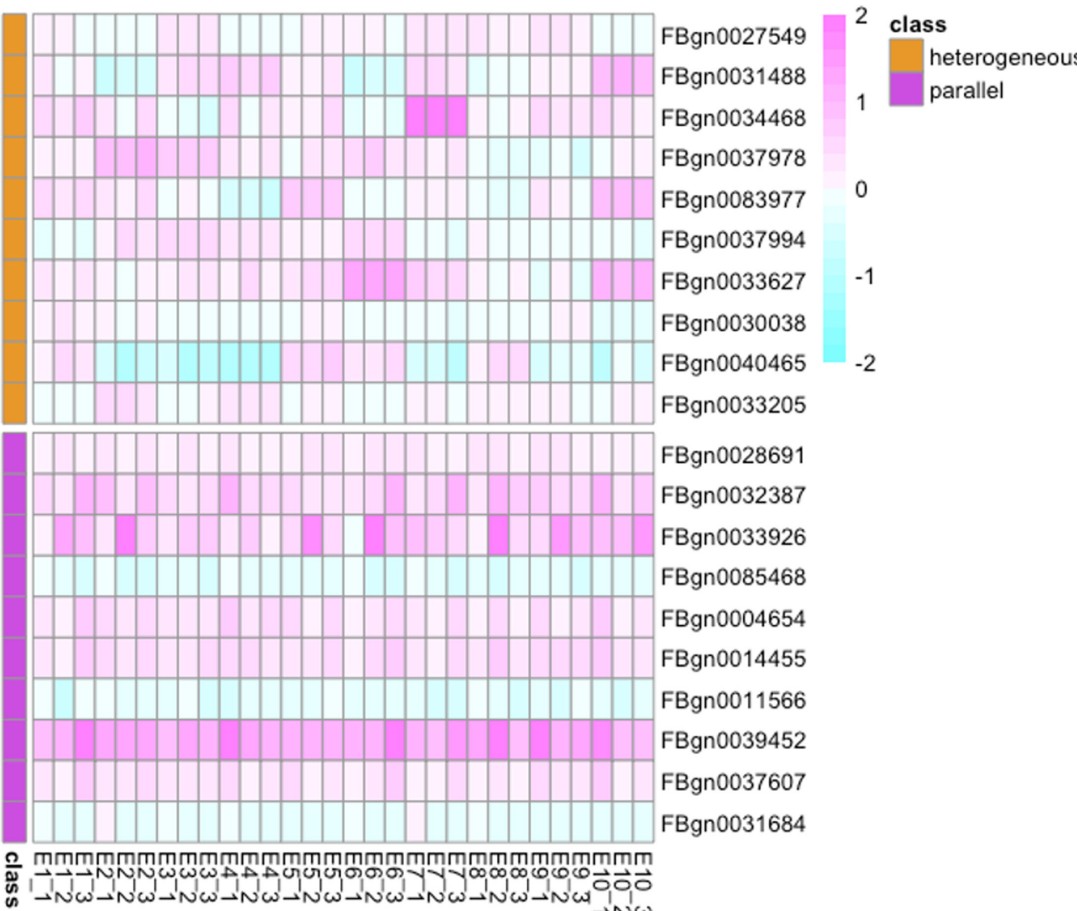

**Appendix 1—figure 2.** Evolutionary changes of gene expression across 10 evolved populations. The figure compares the log$_2$FC of the top 10 genes with the most parallel evolutionary responses (genes with the top 10 highest 1 /F values, shown in pink) and the top 10 genes with the most heterogeneous evolutionary responses (genes with the lowest 1 /F values, shown in orange). As illustrated in the figure, genes with higher 1 /F values display more parallel evolutionary responses (log$_2$FC) across the 10 evolved populations, while genes with lower 1 /F values exhibit more variation in their evolutionary responses.

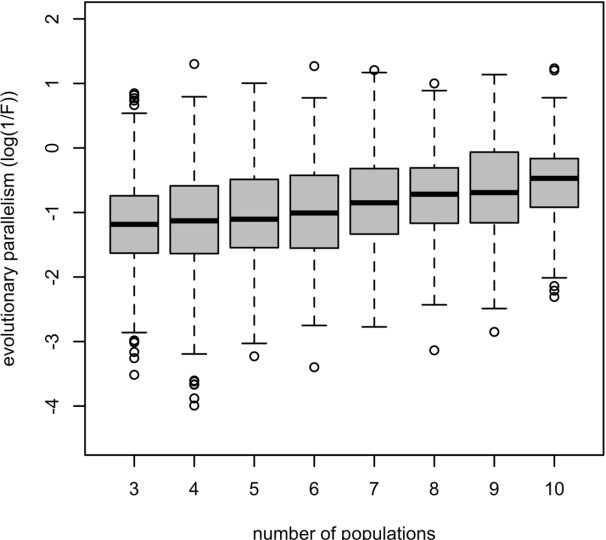

**Appendix 1—figure 3.** Association between parallelism of a gene (log(1 /F); y-axis) and the number of evolved populations in which it is significantly differentiated from the ancestral population (x-axis). A significant positive correlation between them (rho = 0.22, p-value <5.7e-10) suggests that genes being detected in more evolved populations have higher expression parallelism and vice versa.

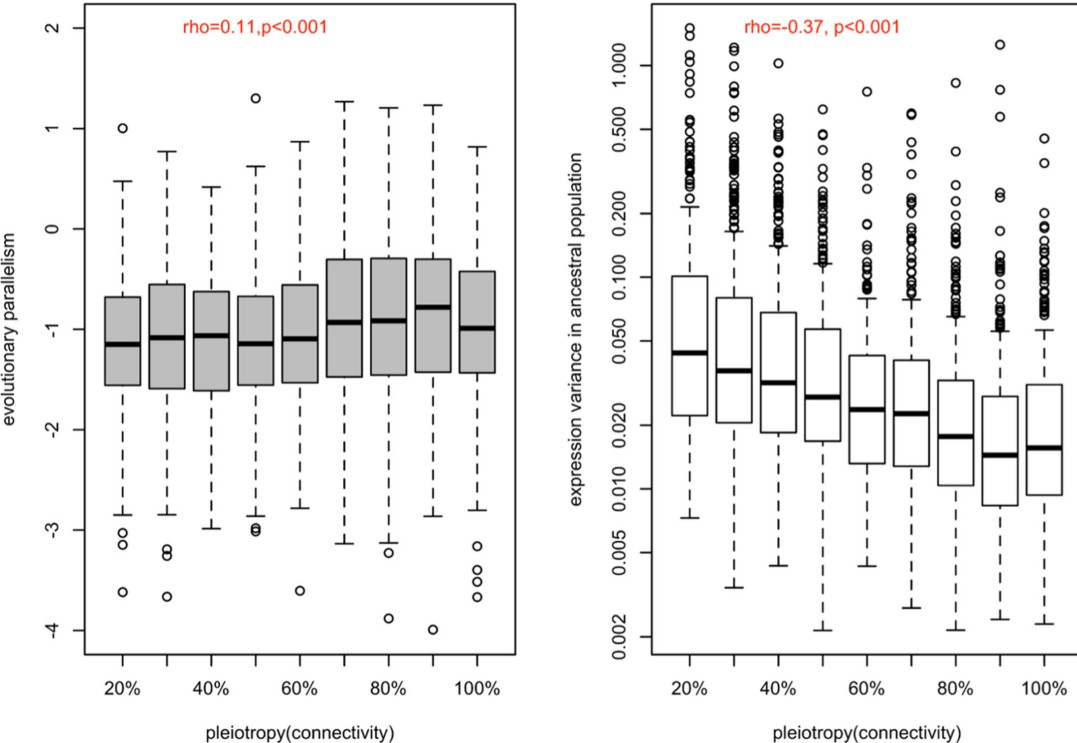

**Appendix 1—figure 4.** Correlation between the strength of pleiotropy (network connectivity) with the evolutionary parallelism (**a**) and with the ancestral variation (**b**). (**a**) The distribution of evolutionary parallelism of different genes is shown in boxplots binned by their strength of pleiotropy (network connectivity). The strength of pleiotropy was positively correlated with evolutionary parallelism (rho = 0.11, p-value <5.6e-07). (**b**) The distribution of expression variance in ancestral populations of different genes is shown in boxplots binned by their strength of pleiotropy. The strength of pleiotropy is negatively correlated with the ancestral variation in gene expression (rho = −0.37, p-value <2.2e-16). Both results were consistent with the finding of the other measure of pleiotropy, tissue specificity.

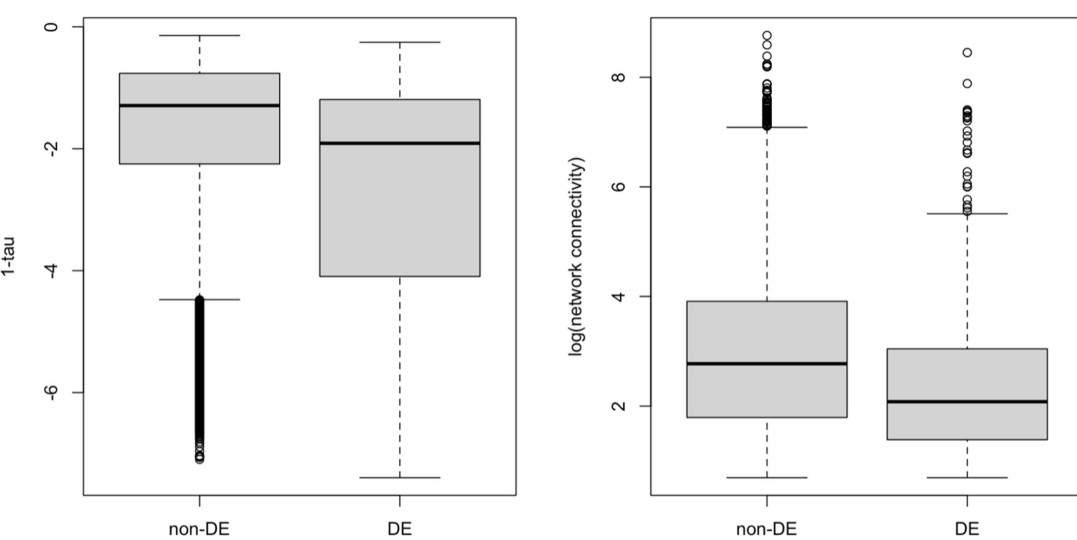

**Appendix 1—figure 5.** Strength of pleiotropy between DE and non-DE gene groups using 1-tissue specificity (**a**) and network connectivity (**b**) as proxies, respectively. Genes without an evolutionary response (non-DE) exhibit greater pleiotropic strength compared to genes with evolutionary responses (differential expression, DE), defined here as genes showing significant expression changes between ancestral and evolved populations in the same direction in at least three populations. This pattern is consistent when using both tissue specificity and network connectivity as measures of pleiotropy, suggesting that extremely high levels of pleiotropy may constrain evolutionary responses.

**Appendix 1—table 1.** Regression of evolutionary parallelism (response) on both measures of pleiotropy (1-$\tau$ and network connectivity) and ancestral variation.

Coefficient estimates and hypothesis test ($\beta = 0$) results are provided.

| Variable | $\beta$ | F-statistics | p-value |
|---|---|---|---|
| Ancestral variation | –0.16 | 76.7227 | <2.2e-16*** |
| 1-$\tau$ | 0.13 | 26.4172 | 3.055e-07*** |
| Network connectivity | –0.03 | 1.1288 | 0.2882 |

**Appendix 1—table 2.** Regression of ancestral variation (response) on both measures of pleiotropy (1-$\tau$ and network connectivity).

Coefficient estimates and hypothesis test ($\beta = 0$) results are provided.

| Variable | $\beta$ | F-statistics | p-value |
|---|---|---|---|
| 1-$\tau$ | –0.236 | 258.581 | <2.2e-16*** |
| Network connectivity | –0.239 | 95.079 | <2.2e-16*** |

