## [Editor Report · eLife Assessment]

This study makes the **important** finding that pleiotropy is positively associated with parallelism of evolutionary responses in gene expression. This finding, if true, runs counter to current expectations in the field. The analysis uses state-of-the art experimental evolution approach to study the genetic basis of adaptation of *Drosophila* simulans to a hot environment. Although the experimental results are **convincing**, the theoretical model is **incomplete**, due to several unusual assumptions. It remains to be seen whether the main conclusion can be replicated in other contexts.

---

## [Referee Report · Reviewer #1 (Public review)]

When different groups (populations, species) are presented with similar environmental pressures, how similar are the ultimate targets (genes, pathways)? This study sought to illuminate this broader question via experimental evolution in D. simulans and quantifying gene-expression changes, specifically in the context of standing genetic variation (and not de novo mutation). Ultimately, the authors showed pleiotropy and standing-genetic variation play a significant role in the "predictability" of evolution.

The results of this manuscript look at the interplay between pleiotropy, standing genetic variation and parallelism (i.e. predictability of evolution) in gene expression. Ultimately, their results suggest that (a) pleiotropic genes typically have a smaller range in variation/expression, and (b) adaptation to similar environments tends to favor changes in pleiotropic genes, which leads to parallelism in mechanisms (though not dramatically). However, it is still uncertain how much parallelism is directly due to pleiotropy, instead of a complex interplay between them and ancestral variation.

---

## [Referee Report · Reviewer #2 (Public review)]

Summary:

Lai and collaborators use a previously published RNAseq dataset derived from an experimental evolution set up to compare the pleiotropic properties of genes which expression evolved in response to fluctuating temperature for over 100 generations. The authors correlate gene pleiotropy with the degree of parallelisms in the experimental evolution set up to ask: are genes that evolved in multiple replicates more or less pleiotropic?

They find that, maybe counter to expectation, highly pleiotropic genes show more replicated evolution. And such effect seems to be driven by direct effects (which the authors can only speculate on) and indirect effect through low variance in pleiotropic genes (which the authors indirectly link to genetic variation underlying gene expression variance).

Weaknesses:

The results offer new insights into the evolution of gene expression and into the parameters that constrain such evolution, i.e., pleiotropy. Although the conclusions are supported by the data, I find the interpretation of the results a little bit complicated.

Major comment:

The major point I ask the authors to address is whether the connection between polygenic adaptation and parallelism can indeed be used to interpret gene expression parallelism. If the answer is not, please rephrase the introduction and discussion, if the answer is yes, please make it explicit in the text why it is so.

The authors argument: parallelism in gene expression is the same as parallelism in SNP allele frequency (AFC) (see L389-383 here they don't mention that this explanation is derived from SNP parallelism and not trait parallelism, and see Fig1 b). In previous publications the authors have explained the low level of AFC parallelism using a polygenic argument. Polygenic traits can reach a new trait optimum via multiple SNPs and therefore although the trait is parallel across replicates, the SNPs are not necessarily so.

In the current paper, they seem to be exchanging SNP AFC by gene expression, and to me, those are two levels that cannot be interchanged. Gene expression is a trait, not a SNP, and therefore the fact that a gene expression doesn't replicate cannot be explained by polygenic basis, because again the trait is gene expression itself. And, actually the results of the simulations show that high polygenicity = less trait parallelism (Fig4).

Now, if the authors focus on high parallel genes (present in e.g. 7 or more replicates) and they show that the eQTLs for those genes are many (highly polygenic) and the AFC of those eQTL are not parallel, then I would agree with the interpretation. But, given that here they just assess gene expression and not eQTL AFC, I do not think they can use the 'highly polygenic = low parallelism' explanation.

The interpretation of the results to me, should be limited to: genes with low variance and high pleiotropy tend to be more parallel, and the explanation might be synergistic pleiotropy.

Comments on revisions: The authors didn't really address any of the comments made by any of the reviewers - basically nothing was changed in the main text. Therefore, I leave my original review unchanged.

---

## [Referee Report · Reviewer #3 (Public review)]

The authors aim to understand how gene pleiotropy affects parallel evolutionary changes among independent replicates of adaptation to a new hot environment of a set of experimental lines of *Drosophila* simulans using experimental evolution. The flies were RNAsequenced after more than 100 generations of lab adaptation and the changes in average gene expression were obtained relative to ancestral expression levels from reconstructed ancestral lines. Parallelism of gene expression change among lines is evaluated as variance in differential gene expression among lines relative to error variance. Similarly, the authors ask how the standing variation in gene expression estimated from a handful of flies from a reconstructed outbred line affects parallelism. The main findings are that parallelism in gene expression responses is positively associated with pleiotropy and negatively associated with expression variation. Those results are in contradiction with theoretical predictions and empirical findings. To explain those seemingly contradictory results the authors invoke the role of synergistic pleiotropy and correlated selection, although they do not attempt to measure either.

Strengths:

The study uses highly replicated outbred laboratory lines of *Drosophila* simulans evolved in the lab under constant hot regime for over 100 generations. This allows for robust comparisons of evolutionary responses among lines.

The manuscript is well written and the hypotheses are clearly delineated at the onset.

The authors have run a causal analysis to understand the causal dependencies between pleiotropy and expression variation on parallelism.

The use of whole-body RNA extraction to study gene expression variation is well justified.

Weaknesses:

The accuracy of the estimate of ancestral phenotypic variation in gene expression is likely low because estimated from a small sample of 20 males from a reconstructed outbred line. It might not constitute a robust estimate of the genetic variation of the evolved lines under study.

There are no estimates of the standing genetic variation of expression levels of the genes under study, only estimates of their phenotypic variation. I wished the authors had been clear about that limitation and had refrained from equating phenotypic variation in expression level with standing genetic variation.

Moreover, since the phenotype studied is gene expression, its genetic basis extends beyond expressed sequences. The phenotypic variation of a gene's expression may thus likely misrepresent the genetic variation available for its evolution. The authors do not present evidence that sequence variation correlates with expression variation.

The authors have not attempted to estimate synergistic pleiotropy among genes, nor how selection acts on gene expression modules. It makes their conclusion regarding the role of synergistic pleiotropy rather speculative.

---

## [Author Response]

The following is the authors’ response to the current reviews.

**Reviewer #1 (Public review):**
When different groups (populations, species) are presented with similar environmental pressures, how similar are the ultimate targets (genes, pathways)? This study sought to illuminate this broader question via experimental evolution in D. simulans and quantifying gene-expression changes, specifically in the context of standing genetic variation (and not de novo mutation). Ultimately, the authors showed pleiotropy and standing-genetic variation play a significant role in the "predictability" of evolution.The results of this manuscript look at the interplay between pleiotropy, standing genetic variation and parallelism (i.e. predictability of evolution) in gene expression. Ultimately, their results suggest that (a) pleiotropic genes typically have a smaller range in variation/expression, and (b) adaptation to similar environments tends to favor changes in pleiotropic genes, which leads to parallelism in mechanisms (though not dramatically). However, it is still uncertain how much parallelism is directly due to pleiotropy, instead of a complex interplay between them and ancestral variation.

Yes, the reviewer is correct that our results for the direct effects of pleiotropy were not consistent for both measures of pleiotropy. We highlight this in the discussion:” Only tissue specificity had a significant direct effect, which was even larger than the indirect effect (Table 2). No significant direct effect was found for network connectivity. The discrepancy between the two measures of pleiotropy is particularly interesting given their significant correlation (Supplementary Figure 1). This suggests that both measures capture aspects of pleiotropy that differ in their biological implications.”

**Reviewer #2 (Public review):**
Summary:Lai and collaborators use a previously published RNAseq dataset derived from an experimental evolution set up to compare the pleiotropic properties of genes which expression evolved in response to fluctuating temperature for over 100 generations. The authors correlate gene pleiotropy with the degree of parallelisms in the experimental evolution set up to ask: are genes that evolved in multiple replicates more or less pleiotropic?They find that, maybe counter to expectation, highly pleiotropic genes show more replicated evolution. And such effect seems to be driven by direct effects (which the authors can only speculate on) and indirect effect through low variance in pleiotropic genes (which the authors indirectly link to genetic variation underlying gene expression variance).Weaknesses:The results offer new insights into the evolution of gene expression and into the parameters that constrain such evolution, i.e., pleiotropy. Although the conclusions are supported by the data, I find the interpretation of the results a little bit complicated.

We are very happy to read that the reviewer finds our conclusions to be supported by the data.

Major comment:The major point I ask the authors to address is whether the connection between polygenic adaptation and parallelism can indeed be used to interpret gene expression parallelism. If the answer is not, please rephrase the introduction and discussion, if the answer is yes, please make it explicit in the text why it is so.

Yes, we think that gene expression parallelism can be explained by polygenic adaptation.

The authors argument: parallelism in gene expression is the same as parallelism in SNP allele frequency (AFC) (see L389-383 here they don't mention that this explanation is derived from SNP parallelism and not trait parallelism, and see Fig1 b). In previous publications the authors have explained the low level of AFC parallelism using a polygenic argument. Polygenic traits can reach a new trait optimum via multiple SNPs and therefore although the trait is parallel across replicates, the SNPs are not necessarily so.In the current paper, they seem to be exchanging SNP AFC by gene expression, and to me, those are two levels that cannot be interchanged. Gene expression is a trait, not a SNP, and therefore the fact that a gene expression doesn't replicate cannot be explained by polygenic basis, because again the trait is gene expression itself. And, actually the results of the simulations show that high polygenicity = less trait parallelism (Fig4).

We agree with the reviewer that it is important to consider different hierarchies when talking about the implications of polygenic adaptation. The lowest hierarchical level is SNP variation and the highest level is fitness. In-between these extreme hierarchical levels is gene expression. While gene expression is a trait itself, as correctly pointed out by the reviewer, it is possible that selection is not favoring a specific trait value, because selection targets a trait on a higher hierarchical level. This implies that not only SNPs, but also intermediate traits such as gene expression can exhibit redundancy. Considering a simple example of one selected trait (e.g. body size), which is affected by the expression level of two genes A and B, each regulated by SNP A1, A2 and B1, B2. It is now possible to modulate the focal trait by allele frequency changes of A1, which in turn will only affect gene A. Alternatively, SNP B2 may change, modifying the expression of gene B, leading to the same change in body size. Hence, we could have redundancy both at the SNP level as well as on the gene expression level (although higher redundancy is expected on the SNP level). Most importantly, this redundancy at intermediate hierarchical levels is not pure theory, but it is supported by empirical evidence. We have shown that redundancy exists not only for gene expression (10.1111/mec.16274) but also for metabolite concentrations (10.1093/gbe/evad098).

Now, if the authors focus on high parallel genes (present in e.g. 7 or more replicates) and they show that the eQTLs for those genes are many (highly polygenic) and the AFC of those eQTL are not parallel, then I would agree with the interpretation. But, given that here they just assess gene expression and not eQTL AFC, I do not think they can use the 'highly polygenic = low parallelism' explanation.

This is clearly an interesting proposed research project, but we doubt that it would result in the expected outcome. Since most of the adaptive gene expression changes are not having a simple genetic basis (10.1093/gbe/evae077) and most expression variation is determined by trans-regulatory effects (10.1038/s41576-020-00304-w), eQTL mapping will most likely not identify all contributing loci. Large effect loci are more easily identified, but they are also expected to be more parallel.

The interpretation of the results to me, should be limited to: genes with low variance and high pleiotropy tend to be more parallel, and the explanation might be synergistic pleiotropy.

We thank the reviewer for the suggestion, but prefer to stick to our interpretation of the data.

Comments on revisions: The authors didn't really address any of the comments made by any of the reviewers - basically nothing was changed in the main text. Therefore, I leave my original review unchanged.

We modestly disagree, in our point to point reply, we respond to all reviewers’ comments. Since, we did not identify any major problem in our manuscript, we only modified the wording in some parts where we felt that a clarification could resolve the misunderstanding of the reviewers. In response to the reviewers’ comments, we added a new paragraph in the discussion and generated a new figure.

**Reviewer #3 (Public review):**
The authors aim to understand how gene pleiotropy affects parallel evolutionary changes among independent replicates of adaptation to a new hot environment of a set of experimental lines of *Drosophila* simulans using experimental evolution. The flies were RNAsequenced after more than 100 generations of lab adaptation and the changes in average gene expression were obtained relative to ancestral expression levels from reconstructed ancestral lines. Parallelism of gene expression change among lines is evaluated as variance in differential gene expression among lines relative to error variance. Similarly, the authors ask how the standing variation in gene expression estimated from a handful of flies from a reconstructed outbred line affects parallelism. The main findings are that parallelism in gene expression responses is positively associated with pleiotropy and negatively associated with expression variation. Those results are in contradiction with theoretical predictions and empirical findings. To explain those seemingly contradictory results the authors invoke the role of synergistic pleiotropy and correlated selection, although they do not attempt to measure either.Strengths:The study uses highly replicated outbred laboratory lines of *Drosophila* simulans evolved in the lab under constant hot regime for over 100 generations. This allows for robust comparisons of evolutionary responses among lines.The manuscript is well written and the hypotheses are clearly delineated at the onset.The authors have run a causal analysis to understand the causal dependencies between pleiotropy and expression variation on parallelism.The use of whole-body RNA extraction to study gene expression variation is well justified.Weaknesses:The accuracy of the estimate of ancestral phenotypic variation in gene expression is likely low because estimated from a small sample of 20 males from a reconstructed outbred line. It might not constitute a robust estimate of the genetic variation of the evolved lines under study.

We agree with the reviewer that variation estimates based on 20 samples are not very precise. Nevertheless, we demonstrated that the estimated variance in gene expression was highly correlated between two independent samples from the same ancestral population. Furthermore, we identified a significant correlation of expression variance with evolutionary parallelism. In other words, the biological signal has been sufficiently strong despite the variance estimate has been noisy.

There are no estimates of the standing genetic variation of expression levels of the genes under study, only estimates of their phenotypic variation. I wished the authors had been clear about that limitation and had refrained from equating phenotypic variation in expression level with standing genetic variation.

The reviewer is right that we did not estimate genetic variation of gene expression, but use expression variation as a proxy for the standing genetic variation. There are two potential problems with this approach. First, a large expression variation could be caused by a single large effect variant segregating at intermediate frequency. Such large effect variants will exhibit a highly parallel selection response-contrary to our empirical results. Since we have shown previously (10.1093/gbe/evae077) that adaptive gene expression changes are mostly polygenic we do not consider this extreme scenario to be very relevant in our study. Rather, we would like to emphasize that neither a SNP analysis of the 5’ region nor an eQTL study will provide an unbiased estimator of genetic variation of gene expression. The second problem arises if gene expression noise differs among genes, hence more noisy genes will appear to have more standing genetic variation than genes with less noise. Since, we average across many different cells and cell types, gene expression noise is expected to be levelled out- this aspect is discussed in detail in the manuscript.

In other words, despite these two potential limitations, we consider our approach superior to alternative approaches of estimating genetic variation in gene expression.

Moreover, since the phenotype studied is gene expression, its genetic basis extends beyond expressed sequences. The phenotypic variation of a gene's expression may thus likely misrepresent the genetic variation available for its evolution. The authors do not present evidence that sequence variation correlates with expression variation.

Gene expression is determined by the joint effects of cis-regulatory and trans-regulatory variation. Hence, recombination can create more extreme phenotypes than the one of the parental lines (in quantitative genetics this is called transgressive segregation). It is unclear to what extent this constitutes a problem for our analyses. Nevertheless, we would like to point out that eQTL mapping will miss many trans-acting variants and therefore we doubt that the requested empirical evidence for correlation between genetic variation (estimated by eQTL mapping) and observed expression variation is as straight forward as suggested by the reviewer.

Nevertheless, we reference an empirical study, which showed a positive correlation between expression variation and cis-regulatory variation.

The authors have not attempted to estimate synergistic pleiotropy among genes, nor how selection acts on gene expression modules. It makes their conclusion regarding the role of synergistic pleiotropy rather speculative.

The reviewer is correct that we did not demonstrate synergistic pleiotropy, but we discuss this as a possible explanation for the observed direct effects of pleiotropy.

The following is the authors’ response to the original reviews.

**Reviewer #1 (Public review):**
The results of this manuscript look at the interplay between pleiotropy, standing genetic variation, and parallelism (i.e. predictability of evolution) in gene expression. Ultimately, their results suggest that (a) pleiotropic genes typically have a smaller range in variation/expression, and (b) adaptation to similar environments tends to favor changes in pleiotropic genes, which leads to parallelism in mechanisms (though not dramatically). However, it is still uncertain how much parallelism is directly due to pleiotropy, instead of a complex interplay between them and ancestral variation.I have a few things that I was uncertain about. It may be these things are easily answered but require more discussion or clarity in the manuscript.(1) The variation being talked about in this manuscript is expression levels, and not SNPs within coding regions (or elsewhere). The cause of any specific gene having a change in expression can obviously be varied - transcription factors, repressors, promoter region variation, etc. Is this taken into account within the "network connectivity" measurement? I understand the network connectivity is a proxy for pleiotropy - what I'm asking is, conceptually, what can be said about how/why those highly pleiotropic genes have a change (or not) in expression. This might be a question for another project/paper, but it feels like a next step worth mentioning somewhere.

In current study, we are only able to detect significant and repeatable expression changes but unable to identify the underlying causal variants. An eQTL study in the founder population in combination with genomic resequencing for both evolved and ancestral populations would be required to address this question.

(2) The authors do have a passing statement in line 361 about cis-regulatory regions. Is the assumption that genetic variation in promoter regions is the ultimate "mechanism" driving any change in expression? In the same vein, the authors bring up a potential confounding factor, though they dismiss it based on a specific citation (lines 476-481; citation 65). I'm of the mindset that in order to more confidently disregard this "issue" based on previous evidence, it requires more than one citation. Especially since the one citation is a plant. That specific point jumps out to me as needing a more careful rebuttal.

It was not our intention to claim that the expression changes in our experiment are caused by cis-regulatory variation only. We believe that the observed expression variation has both cis- and trans-genetic components, where as some studies tend to estimate much higher cisvariation for gene expression in *Drosophila* populations (e.g. [1, 2]). We mentioned the positive correlation between cis-regulatory polymorphism and expression variation to (1) highlight the genetic control of gene expression and (2) make the connection between polygenic adaptation and gene expression evolutionary parallelism.

(3) I feel like there isn't enough exploration of tissue specificity versus network connectivity. Tissue specificity was best explained by a model in which pleiotropy had both direct and indirect effects on parallelism; while network connectivity was best explained (by a small margin) via the model which was mostly pleiotropy having a direct effect on ancestral variation, that then had a direct effect on parallelism. When the strengths of either direct/indirect effects were quantified, tissue specificity showed a stronger direct effect, while network connectivity had none (i.e. not significant). My confusion is with the last point - if network connectivity is explained by a direct effect in the best-supported model, how does this work, since the direct effect isn't significant? Perhaps I am misunderstanding something.

To clarify, for network connectivity, there’s a significant “indirect” effect on parallelism (i.e. network connectivity affect ancestral gene expression and ancestral gene expression affect parallelism). Hence, in table 2, the direct effect of network connectivity on parallelism is weak and not significant while the indirect effect via ancestral variation is significant.

Also, network connectivity might favor the most pleiotropic genes being transcription factor hubs (or master regulators for various homeostasis pathways); while the tissue specificity metric perhaps is a kind of a space/time element. I get that a gene having expression across multiple tissues does fit the definition of pleiotropy in the broad sense, but I'm wondering if some important details are getting lost - I'm just thinking about the relative importance of what tissue specificity measurements say versus the network connectivity measurement.

We examined the statistical relationship between the two measures and found a moderate positive correlation on the basis of which we argued that the two measures may capture different aspects of pleiotropy. We appreciate the reviewer’s suggestions about the biological basis of the two estimates of pleiotropy, but we think that without further experimental insights, an extended discussion of this topic is too premature to provide meaningful insights to the readership.

**Reviewer #2 (Public review):**
Summary:Lai and collaborators use a previously published RNAseq dataset derived from an experimental evolution set up to compare the pleiotropic properties of genes whose expression evolved in response to fluctuating temperature for over 100 generations. The authors correlate gene pleiotropy with the degree of parallelisms in the experimental evolution set up to ask: are genes that evolved in multiple replicates more or less pleiotropic?They find that, maybe counter to expectation, highly pleiotropic genes show more replicated evolution. Such an effect seems to be driven by direct effects (which the authors can only speculate on) and indirect effects through low variance in pleiotropic genes (which the authors indirectly link to genetic variation underlying gene expression variance).Weaknesses:The results offer new insights into the evolution of gene expression and into the parameters that constrain such evolution, i.e., pleiotropy. Although the conclusions are supported by the data, I find the interpretation of the results a little bit complicated.Major comment:The major point I ask the authors to address is whether the connection between polygenic adaptation and parallelism can indeed be used to interpret gene expression parallelism. If the answer is not, please rephrase the introduction and discussion, if the answer is yes, please make it explicit in the text why it is so.

Our answer is yes, we interpreted gene expression parallelism (high ancestral variance -> less parallelism) using the same framework that links polygenic adaptation and parallelism (high polygenicity = less trait parallelism). We believe that our response covers several of the reviewer’s concerns.

The authors' argument: parallelism in gene expression is the same as parallelism in SNP allele frequency (AFC) (see L389-383 here they don't mention that this explanation is derived from SNP parallelism and not trait parallelism, and see Figure 1 b). In previous publications, the authors have explained the low level of AFC parallelism using a polygenic argument. Polygenic traits can reach a new trait optimum via multiple SNPs and therefore although the trait is parallel across replicates, the SNPs are not necessarily so.

Importantly, our rationale is based on the idea that gene expression is rarely the direct target of selection, but rather an intermediate trait [3]. Recently, we have specifically tested this assumption for gene expression and metabolite concentrations and our analysis showed that both traits were are redundant [4], as previously shown for DNA sequences [5]. The important implication for this manuscript is that gene expression is also redundant, so that adaptation can be achieved by distinct changes in gene expression in replicate populations adapting to the same selection pressure. This implies that we can use the same simulation framework for gene expression as for sequencing data. In our case different SNP frequencies correspond to different expression levels (averaged across individuals from a population), which in turn increases fitness by modifying the selected trait. Importantly, the selected trait in our simulations is not gene expression, but a not defined high level phenotype. A key insight from our simulations is that with increasing polygenicity the expression of a gene is more variable in the ancestral population.

In the current paper, they seem to be exchanging SNP AFC by gene expression, and to me, those are two levels that cannot be interchanged. Gene expression is a trait, not an SNP, and therefore the fact that a gene expression doesn't replicate cannot be explained by a polygenic basis, because again the trait is gene expression itself. And, actually, the results of the simulations show that high polygenicity = less trait parallelism (Figure 4).

As detailed above, because adaptation can be reached by changes in gene expression at different sets of genes, redundancy is also operating on the expression level not just on the level of SNPs. To clarify, the x-axis of Fig. 4 is the expression variation in the ancestral population.

Now, if the authors focus on high parallel genes (present in e.g. 7 or more replicates) and they show that the eQTLs for those genes are many (highly polygenic) and the AFC of those eQTLs are not parallel, then I would agree with the interpretation. But, given that here they just assess gene expression and not eQTL AFC, I do not think they can use the 'highly polygenic = low parallelism' explanation.The interpretation of the results to me, should be limited to: genes with low variance and high pleiotropy tend to be more parallel, and the explanation might be synergistic pleiotropy.

While we understand the desire to model the full hierarchy from eQTLs to gene expression and adaptive traits, we raise caution that this would be a very challenging task. eQTLs very often underestimate the contribution of trans-acting factors, hence the understanding of gene expression evolution based on eQTLs is very likely incomplete and cannot explain the redundancy of gene expression during adaptation. Hence, we think that the focus on redundant gene expression is conceptually simpler and thus allows us to address the question of pleiotropy without the incorporation of allele frequency changes.

**Reviewer #3 (Public review):**
The authors aim to understand how gene pleiotropy affects parallel evolutionary changes among independent replicates of adaptation to a new hot environment of a set of experimental lines of *Drosophila* simulans using experimental evolution. The flies were RNAsequenced after more than 100 generations of lab adaptation and the changes in average gene expression were obtained relative to ancestral expression levels from reconstructed ancestral lines. Parallelism of gene expression change among lines is evaluated as variance in differential gene expression among lines relative to error variance. Similarly, the authors ask how the standing variation in gene expression estimated from a handful of flies from a reconstructed outbred line affects parallelism. The main findings are that parallelism in gene expression responses is positively associated with pleiotropy and negatively associated with expression variation. Those results are in contradiction with theoretical predictions and empirical findings. To explain those seemingly contradictory results the authors invoke the role of synergistic pleiotropy and correlated selection, although they do not attempt to measure either.Strengths:(1) The study uses highly replicated outbred laboratory lines of *Drosophila* simulans evolved in the lab under a constant hot regime for over 100 generations. This allows for robust comparisons of evolutionary responses among lines.(2) The manuscript is well written and the hypotheses are clearly delineated at the onset.(3) The authors have run a causal analysis to understand the causal dependencies between pleiotropy and expression variation on parallelism.(4) The use of whole-body RNA extraction to study gene expression variation is well justified.Weaknesses:(1) It is unclear how well phenotypic variation in gene expression of the evolved lines has been estimated by the sample of 20 males from a reconstructed outbred line not directly linked to the evolved lines under study. I see this as a general weakness of the experimental design.

Our intention was not to measure the phenotypic variance of the evolved lines, but rather to estimate the phenotypic variance at the beginning of the experiment. Hence, we measured and investigated the variation of gene expression in the ancestral population since this was the beginning of the replicated experimental evolution. Furthermore, since the ancestral population represents the natural population in Florida, the gene expression variation reflects the history of selection history acting on it.

(2) There are no estimates of standing genetic variation of expression levels of the genes under study, only phenotypic variation. I wished the authors had been clear about that limitation and had discussed the consequences of the analysis. This also constitutes a weakness of the study.

The reviewer is correct that we do not aim to estimate the standing genetic variation, which is responsible for differences in gene expression. While we agree that it could be an interesting research question to use eQTL mapping to identify the genetic basis of gene expression, we caution that trans-effects are difficult to estimate and therefore an important component of gene expression evolution will be difficult to estimate. Hence, we consider that our focus on variation in gene expression without explicit information about the genetic basis is simpler and sufficient to address the question about the role of pleiotropy.

(3) Moreover, since the phenotype studied is gene expression, its genetic basis extends beyond expressed sequences. The phenotypic variation of a gene's expression may thus likely misrepresent the genetic variation available for its evolution. The genetic variation of gene expression phenotypes could be estimated from a cross or pedigree information but since individuals were pool-sequenced (by batches of 50 males), this type of analysis is not possible in this study.

We agree with the reviewer that gene expression variation may also have a non-genetic basis, we discuss this in depth in the discussion of the manuscript.

(4) The authors have not attempted to estimate synergistic pleiotropy among genes, nor how selection acts on gene expression modules. It makes any conclusion regarding the role of synergistic pleiotropy highly speculative.

We mentioned synergistic pleiotropy as a possible explanation for our results. A positive correlation between the fitness effect of gene expression variation would predict more replicable evolutionary changes. A similar argument has been made by [6].

I don't understand the reason why the analysis would be restricted to significantly differentially expressed genes only. It is then unclear whether pleiotropy, parallelism, and expression variation do play a role in adaptation because the two groups of adaptive and non-adaptive genes have not been compared. I recommend performing those comparisons to help us better understand how "adaptive" genes differentially contribute to adaptation relative to "nonadaptive" genes relative to their difference in population and genetic properties.

We agree with the reviewer that the comparison between the pleiotropy of adaptive and nonadaptive genes is interesting. We performed the analysis but omitted from the current manuscript for simplicity. Similar to the results in [6], non-adaptive genes are more pleiotropic than the adaptive genes. For adaptive genes we find a positive correlation between the level of pleiotropy and evolutionary parallelism. Thus, high pleiotropy limits the evolvability of a gene, but moderate and potentially synergistic pleiotropy increases the repeatability of adaptive evolution. We included this result in the revised manuscript and discuss it.

There is a lack of theoretical groundings on the role of so-called synergistic pleiotropy for parallel genetic evolution. The Discussion does not address this particular prediction. It could be removed from the Introduction.

We modestly disagree with the reviewer, synergistic pleiotropy is covered by theory and empirical results also support the importance of synergistic pleiotropy.

References

(1) Genissel A, McIntyre LM, Wayne ML, Nuzhdin SV. Cis and trans regulatory effects contribute to natural variation in transcriptome of *Drosophila melanogaster*. Molecular biology and evolution. 2008;25(1):101-10. Epub 20071112. doi: 10.1093/molbev/msm247. PubMed PMID: 17998255.

(2) Osada N, Miyagi R, Takahashi A. Cis- and Trans-regulatory Effects on Gene Expression in a Natural Population of *Drosophila melanogaster*. Genetics. 2017;206(4):2139-48. Epub 20170614. doi: 10.1534/genetics.117.201459. PubMed PMID: 28615283; PubMed Central PMCID: PMCPMC5560811.

(3) Barghi N, Hermisson J, Schlötterer C. Polygenic adaptation: a unifying framework to understand positive selection. Nature reviews Genetics. 2020;21(12):769-81. Epub 2020/07/01. doi: 10.1038/s41576-020-0250-z. PubMed PMID: 32601318.

(4) Lai WY, Otte KA, Schlötterer C. Evolution of Metabolome and Transcriptome Supports a Hierarchical Organization of Adaptive Traits. Genome biology and evolution. 2023;15(6). Epub 2023/05/26. doi: 10.1093/gbe/evad098. PubMed PMID: 37232360; PubMed Central PMCID: PMCPMC10246829.

(5) Barghi N, Tobler R, Nolte V, Jaksic AM, Mallard F, Otte KA, et al. Genetic redundancy fuels polygenic adaptation in *Drosophila*. PLoS biology. 2019;17(2):e3000128. Epub 2019/02/05. doi: 10.1371/journal.pbio.3000128. PubMed PMID: 30716062.

(6) Rennison DJ, Peichel CL. Pleiotropy facilitates parallel adaptation in sticklebacks. Molecular ecology. 2022;31(5):1476-86. Epub 2022/01/09. doi: 10.1111/mec.16335. PubMed PMID: 34997980; PubMed Central PMCID: PMCPMC9306781.